# Expansion of CD10$^{neg}$ neutrophils and CD14$^{+}$HLA-DR$^{neg/low}$ monocytes driving proinflammatory responses in patients with acute myocardial infarction

Daniela Fraccarollo*, Jonas Neuser, Julian Möller, Christian Riehle, Paolo Galuppo, Johann Bauersachs

Department of Cardiology and Angiology, Hannover Medical School, Hannover, Germany

**Abstract** Immature neutrophils and HLA-DR$^{neg/low}$ monocytes expand in cancer, autoimmune diseases and viral infections, but their appearance and immunoregulatory effects on T-cells after acute myocardial infarction (AMI) remain underexplored. We found an expansion of circulating immature CD16$^{+}$CD66b$^{+}$CD10$^{neg}$ neutrophils and CD14$^{+}$HLA-DR$^{neg/low}$ monocytes in AMI patients, correlating with cardiac damage, function and levels of immune-inflammation markers. Immature CD10$^{neg}$ neutrophils expressed high amounts of MMP-9 and S100A9, and displayed resistance to apoptosis. Moreover, we found that increased frequency of CD10$^{neg}$ neutrophils and elevated circulating IFN-γ levels were linked, mainly in patients with expanded CD4$^{+}$CD28$^{null}$ T-cells. Notably, the expansion of circulating CD4$^{+}$CD28$^{null}$ T-cells was associated with cytomegalovirus (CMV) seropositivity. Using bioinformatic tools, we identified a tight relationship among the peripheral expansion of immature CD10$^{neg}$ neutrophils, CMV IgG titers, and circulating levels of IFN-γ and IL-12 in patients with AMI. At a mechanistic level, CD10$^{neg}$ neutrophils enhanced IFN-γ production by CD4$^{+}$ T-cells through a contact-independent mechanism involving IL-12. In vitro experiments also highlighted that HLA-DR$^{neg/low}$ monocytes do not suppress T-cell proliferation but secrete high levels of pro-inflammatory cytokines after differentiation to macrophages and IFN-γ stimulation. Lastly, using a mouse model of AMI, we showed that immature neutrophils (CD11b$^{pos}$Ly6G$^{pos}$CD101$^{neg}$ cells) are recruited to the injured myocardium and migrate to mediastinal lymph nodes shortly after reperfusion. In conclusion, immunoregulatory functions of CD10$^{neg}$ neutrophils play a dynamic role in mechanisms linking myeloid cell compartment dysregulation, Th1-type immune responses and inflammation after AMI.

*For correspondence:
fraccarollo.daniela@mh-hannover.de

Competing interests: The authors declare that no competing interests exist.

## Introduction

Despite advances in interventional therapies patients with large acute myocardial infarction (AMI) are at higher risk of heart failure morbidity and mortality (*Heusch and Gersh, 2017*). Immunity and inflammation play a key role in the pathogenesis of ischemic heart failure, and the complex role of immune cells during the wound healing process after injury is currently the focus of intensive research efforts. Understanding the immune mechanisms operating during AMI could pave the way to develop more effective strategies to prevent progressive dilative cardiac remodeling, functional deterioration and heart failure and to reduce adverse cardiovascular events.

HLA-DR$^{neg/low}$ monocytes and immature neutrophils expand in pathological conditions such as cancer, infection, and inflammation, (*Gabrilovich and Nagaraj, 2009*) and have recently been implicated in the pathogenesis of severe COVID-19 (*Silvin et al., 2020*; *Deutsche COVID-19 OMICS Initiative (DeCOI) et al., 2020*). A rapid depression of monocytic HLA-DR expression was observed in

patients with AMI (*Haeusler et al., 2012*). *Maréchal et al., 2020* studying neutrophil phenotypes in acute coronary syndrome found increased percentage of immature low density neutrophils in patients with ST-elevation MI. However, the role of immature neutrophils as well as HLA-DR[neg/low] monocytes in immune mechanisms operating during AMI remains largely unexplored.

By integrating flow cytometric immunophenotyping of monocyte, neutrophil, and lymphocyte subsets, ex vivo experiments with sorted cells as well as bioinformatic tools this study investigated the appearance and the functional immune properties of immature neutrophils and HLA-DR[neg/low] monocytes in patients with AMI. Moreover, we explored whether increased frequency of immature neutrophils and HLA-DR[neg/low] monocytes are linked to circulating levels of immune regulators and acute inflammation markers such as G-CSF, S100A9/S100A8, MMP-9, NGAL, MPO, IL-6, TNF-α, IL-1ß and IFN-γ.

Since a crucial role for immature neutrophils in the orchestration of adaptive immunity has recently emerged, we mostly focused on neutrophil-mediated regulation of T-cell response. Environmental factors especially cytomegalovirus (CMV) infection are likely to provide a significant contribution to functional diversity of T-cells. Therefore, we also investigated the impact of CMV on CD4[+] T-cell homeostasis and on the immunoregulatory properties of immature neutrophils derived from patients with AMI.

In addition, using a mouse model of reperfused AMI we addressed whether immature neutrophils migrate into the injured myocardium and populate heart-draining lymph nodes in response to acute ischemic injury.

## Methods

### Patients and study design

The study protocol is in accordance with the ethical guidelines of the 1975 declaration of Helsinki and has been approved by the local ethics committee of Hannover Medical School. Patients referred to our department for acute coronary syndrome (ACS) were included after providing written informed consent. Patients suffering from active malignant diseases or receiving immunosuppressive therapy were not included. Seventy-one patients (*Table 1*) were categorized into unstable angina (UA, n=11), Non-ST-elevation MI (NSTEMI, n=16), and ST-elevation MI (STEMI, n=44). Left ventricular (LV) ejection fraction was measured in 2D echocardiographic studies using bi-plane Simpson's method. Seventeen healthy volunteers were recruited as control subjects.

**Table 1.** General traits.

|  |  | UA (*N*=11) | NSTEMI (*N*=16) | STEMI (*N*=44) |
|---|---|---|---|---|
| Age (years) |  | 63.3±2.5 | 64.9±3.4 | 60.6±1.7 |
| Gender | Male/Female | 9/2 | 14/2 | 36/8 |
| BMI (kg/m$^2$) |  | 27.5±0.9 | 28.2±0.9 | 27.5±0.7 |
| Blood analyses | LDL (mg/dL) | 92.2±19.2 | 95.5±9.5 | 138.2±7.3 |
|  | CK (IU/L) | 120.0 (87.0–444.0) | 189.0 (126.8–377.0) | 373.5 (110.5–931.2) |
|  | CK$_{max}$ (IU/L) | 120.0 (86.5–440.0) | 403.5 (150.5–578.2) | 1343.5 (574.8–1917.0) |
|  | CK-MB (IU/L) | 19.0 (17.0–22.0) | 32.0 (23.5–57.0) | 47.0 (24.5–91.5) |
|  | LVEF (%) | 53.3±3.7 | 54.1±2.2 | 51.0±1.5 |
|  | Troponin (ng/L) | 12.8 (5.3–22.7) | 99.0 (36.7–273.5) | 337.0 (84.0–962.0) |
|  | Creatinine (μmol/L) | 83.0±4.5 | 88.6±4.8 | 98.5±8.3 |
|  | CRP (mg/L) | 1.6 (0.7–3.4) | 1.8 (1.1–4.3) | 2.5 (1.2–4.7) |

Data are presented as mean ± SEM or as median (IQR). LDL, low density lipoprotein; CK, creatine kinase; CK$_{max}$, maximum CK; CK-MB, creatine kinase-myocardial band; LVEF, left ventricular ejection fraction; CRP, C-reactive protein.

## Flow cytometry

Venous blood was collected in EDTA tubes, stored at room temperature and processed within 1 hr of collection. White blood cell count was measured by an automated hematology analyzer (XT 2000i, Sysmex). Serum was separated within 45 min and stored at −80°C. For multiparameter flow cytometry whole blood (100 µL) was incubated with fluorochrome-conjugated antibodies for 30 min at room temperature in the dark, followed by lysis of red blood cells with Versalyse Lysing Solution (Beckman Coulter) (*Haghikia et al., 2018*). Finally, the cells were washed twice with Hanks' Balanced Salt solution (4 mL). For cell sorting or flow cytometric analysis of monocyte-derived macrophages, cells were resuspended in ice-cold FACS-staining buffer (PBS, supplemented with 0.5% bovine serum albumin and 2 mM EDTA) and immunostaining was performed on ice. The following antibodies were used: anti-CD14 (Clone M5E2, 1:50, BD Biosciences; RRID:AB_1645464); anti HLA-DR (Clone L243, 1:30, BioLegend; RRID:AB_314682, RRID:AB_314684, RRID:AB_893567); anti-CD16 (Clone 3G8, 1:50, BioLegend; RRID:AB_2562990); anti-CX3CR1 (Clone 2A9-1, 1:50, BioLegend; RRID:AB_1595456); anti-CCR2 (Clone K036C2, 1:50, BioLegend; RRID:AB_2562004); anti-CD66b (Clone G10F5, 1:30, BioLegend; RRID:AB_314496); anti-CD10 (Clone HI10a, 1:20, BioLegend; RRID: AB_2561833, RRID:AB_314914); anti-CD3 (Clone SK7, 1:30, BD Biosciences, RRID:AB_1645475); anti-CD4 (Clone RPA-T4, 1:30, BD Biosciences; RRID:AB_10895807); anti-CD28 (Clone CD28.2, 1:30, BioLegend; RRID:AB_2561910); anti-CCR7 (Clone G043H7, 1:30, BioLegend; RRID:AB_10916389); anti-CD45RA (Clone HI 100, 1:30, BioLegend; RRID:AB_314412); anti-CD114 (Clone LMM741, 1:25, BioLegend; RRID:AB_2083867); anti-CD177 (Clone MEM-166, 1:25, BioLegend; RRID:AB_2072603); anti-CD11b (Clone ICRF44, 1:25, BioLegend; RRID:AB_10933428); anti-CD101 (Clone BB27, 1:25, BioLegend; RRID:AB_2716106); anti-MERTK (Clone 590H11G1E3, 1:25, BioLegend; RRID:AB_2687289); anti-CD163 (Clone GHI/61, 1:25, BD Biosciences; RRID:AB_2738379). Fluorescence minus one (FMO) controls were included during acquisition for gating analyses to distinguish positive from negative staining cell populations. FACS data were acquired on a Gallios flow cytometer and analyzed with Gallios software (Beckman Coulter).

## Isolation of blood mononuclear cells and neutrophils

Peripheral blood was collected in EDTA tubes and mononuclear cells (PBMC) were isolated by density gradient centrifugation using Ficoll-Paque Premium (GE Healthcare Biosciences). CD14$^+$HLA-DR$^{neg/low}$/CD14$^+$HLA-DR$^{high}$ monocytes were FACS-sorted from PBMC. Granulocytes/neutrophils were isolated from the erythrocyte fraction by dextran sedimentation or from whole blood by immunomagnetic selection (130-104-434, MACSxpress Whole Blood Neutrophil Isolation Kit; Miltenyi Biotec). CD10$^{neg}$/CD10$^{pos}$ neutrophils were separated by flow-cytometric sorting. Cells were sorted in RTL Lysis Buffer plus 1% β-mercaptoethanol (74134, RNeasy Plus Mini Kit; QIAGEN), or in sterile Sorting Medium [RPMI 1640 supplemented with 10% (v/v) Heat-Inactivated Fetal Bovine Serum (HI-FCS; A3840001; Gibco) or with 5% autologous serum]. Cell sorting was performed using a FACS Aria Fusion or FACS Aria Ilu (BD Biosciences). Cytospin (Shandon Cytospin 4; Thermo Scientific) preparations were stained with May-Grünwald Giemsa or Wright-Giemsa (Polysciences Europe GmbH; Astral Diagnostics).

## Macrophage generation and stimulation

For in vitro differentiation of monocytes into macrophages, FACS-sorted cells were suspended at $0.5 \times 10^6$ cells/mL in RPMI 1640 medium supplemented with 10% HI-FCS and 1% PenStrep (10378016; Gibco). CD14$^+$HLA-DR$^{neg/low}$/CD14$^+$HLA-DR$^{high}$ monocytes were cultured in 96 well plates (200 µL/well) in the presence of 20 ng/mL M-CSF (216-MC-005; R and D Systems) for 4 days (*Murray et al., 2014*). Monocyte-derived macrophages [(Mb, in RPMI 1640 medium supplemented with 2% HI-FCS] were stimulated with IFN-γ [M(IFN-γ), 20 ng/mL; 285-IF; R and D Systems], IL-4 [M(IL-4), 20 ng/mL, 130-093-920; Miltenyi Biotec], or dexamethasone [M(dexa), 1 µM; Sigma] (*Xue et al., 2014*) for 48 hr.

## T-cell proliferation assays in presence of monocytes

Isolation of CD3$^+$ T-cells was performed using Dynabeads Untouched Human T-cells Kit (11344D, Invitrogen). CD3$^+$ T-cells were stained with CellTrace Violet Cell Proliferation Kit (C34571; Invitrogen) and resuspended at 1×10$^6$/mL in T-Cell Activation Medium (OpTmizer CTS T-Cell Expansion culture

medium supplemented with L-glutamine/PenStrep). CD3$^+$ T-cells were co-cultured in 96-well plates with CD14$^+$HLA-DR$^{neg/low}$ and CD14$^+$HLA-DR$^{high}$ monocytes at a ratio of 1 to 1 (T-cells: monocytes). T-cells were stimulated with Dynabeads Human T-Activator CD3/CD28 (11131D; Gibco) and T-cell proliferation was assessed 4 days later by CellTrace Violet dilution by flow cytometry.

## Chemotaxis assay

Chemotaxis of CD14$^+$HLA-DR$^{neg/low}$/CD14$^+$HLA-DR$^{high}$ monocytes was analyzed with the ChemoTx Disposable Chemotaxis System (NeuroProbe, NRP-106–8; Hölzel Diagnostika) (*Chen et al., 2015*). Human S100A9 (A42590, Invitrogen) was diluted in RPMI 1640 medium and placed in the lower chamber. Cells were washed with RPMI 1640 medium and preincubated for 30 min at 37°C before addition to the upper chamber. Monocytes were allowed to migrate for 3 hr and migration was analyzed by flow cytometry. The migration index was defined as the number of cells migrating in response to S100A9 divided by the number of cells migrating in response to medium alone (*Chen et al., 2015*).

## Assessment of neutrophil apoptosis

CD10$^{neg}$/CD10$^{pos}$ neutrophils (1x10$^6$/mL) were cultured for 24 hr in RPMI 1640 medium supplemented with 5% HI-FCS and 1% PenStrep. Apoptosis rate was assessed by flow cytometry using a Vybrant DyeCycle Violet/SYTOX AADvanced Apoptosis Kit (A35135, Invitrogen) and Apotracker Green (*Barth et al., 2020*) (427402; Biolegend), according to the manufacturer's protocols. Stained neutrophils were analyzed using a Gallios cytometer and Gallios software (Beckman Coulter).

## T-cell activation assays in presence of CD10$^{neg}$/CD10$^{pos}$ neutrophils

CD4$^+$ T-cells were isolated from PBMC using the MojoSort Human CD4 T Cell Isolation Kit (480009; BioLegend) or by flow-cytometric sorting. The CD28 MicroBead Kit (130-093-247; Miltenyi Biotec) was used for isolation of CD4$^+$CD28$^{null}$ T-cells from PBMC. CD4$^+$ T-cells and CD4$^+$CD28$^{null}$ T-cells were resuspended at 1x10$^6$/mL in T-Cell Activation Medium and stimulated with Dynabeads Human T-Activator CD3/CD28. For transwell experiments CD4$^+$ T-cells and CD10$^{neg}$/CD10$^{pos}$ neutrophils were co-cultured in 24 well plates at a ratio of 1 to 2 (T-cells: neutrophils) for 24 hr. CD10$^{neg}$/CD10$^{pos}$ neutrophils were cultured in 0.4 µm transwell inserts (140620, Thermo Scientific) and CD4$^+$ T-cells in the well beneath the insert. In some experiments, CD10$^{neg}$/CD10$^{pos}$ neutrophils were cultured overnight in T-Cell Activation Medium. The cell-free supernatants derived from CD10$^{neg}$/CD10$^{pos}$ neutrophils were added to CD4$^+$ T-cells cultured in 96-well plates (8$\times$10$^4$ cells/well) in the presence of neutralizing anti-IL-12 antibody (4 µg/mL; MAB219, R and D Systems; RRID:AB_2123616) or isotype control (4 µg/mL; MAB002, R and D Systems; RRID:AB_357344). CD4$^+$CD28$^{null}$ T-cells were cultured with cell-free supernatants derived from CD10$^{neg}$ neutrophils. Culture supernatants were collected after 24 hr incubation.

## LEGENDplex and ELISA assays

Blood levels of G-CSF, MMP9, S100A9/S100A8, NGAL, MPO, TNF-$\alpha$, IL-6, IL-1ß, and IFN-$\gamma$ were measured using bead-based multiplex assays (740180; 740589; 740929; LEGENDplex BioLegend). Serum samples were screened for CMV-specific IgG antibodies with the CMV-IgG-ELISA PKS Medac enzyme immunoassay (115-Q-PKS; Medac Diagnostika), using a cut-off value of >0.55 AU/mL for defining seropositivity according to manufacturer's guidelines. Levels of IFN-$\gamma$, IL-12, TNF-$\alpha$, IL-6, and IL-1ß in the cell-culture supernatants were measured by ELISA (DIF50; R and D Systems) and using bead-based immunoassay (740929; LEGENDplex BioLegend).

## RT-quantitative PCR

RNA was isolated from cells sorted in RTL Lysis Buffer using the RNeasy Plus Mini Kit (QIAGEN) according to the manufactures' protocol. RNA quantification and quality testing were assessed by NanoDrop 2000 (Thermo Fisher Scientific) and Bioanalyzer 2100 (Agilent).

cDNA synthesis was performed using 3 ng (human-neutrophils), 1 ng (mouse-neutrophils), and 10 ng (monocytes) of total RNA and iScript Reverse Transcription Supermix (Bio-Rad). Relative quantitation of mRNA expression levels was determined with CFX96 Touch Real Time PCR using SsoAdvanced Universal SYBR Green Supermix and PrimePCR Primers (Bio-Rad). ß-actin (ACTB) was chosen

as an endogenous control. PCR amplification was performed at initially 95℃ for 30 s followed by 40 cycles at 95℃ for 5 s and terminated by 60℃ for 30 s. The delta-delta Ct method was employed for data analysis.

## Animal experiments

### Study protocol

All animal experiments were conducted in accordance with the Guide for the Care and Use of Laboratory Animals published by the National Institutes of Health (Publication No. 85–23, revised 1985). All procedures were approved by the Regierung von Unterfranken (Würzburg, Germany; permit No. 54–2531.01-15/07) and by the Niedersächsisches Landesamt für Verbraucherschutz und Lebensmittelsicherheit (Oldenburg, Germany; permit No. 33.12-42502-04-11/0644; 33.9-42502-04-13/1124 and 33.12-42502-04-17/2702). C57BL/6 mice of both sexes were used in this study (*Fraccarollo et al., 2019*; *Galuppo et al., 2017*; *Fraccarollo et al., 2017*; *Fraccarollo et al., 2011*).

### Mouse model of reperfused AMI

Myocardial ischemia was induced by transient left coronary artery ligation in age- and gender-matched mice. Briefly, mice were anesthetized with 2% isoflurane in a 100% oxygen mix, intubated, and ventilated using a ventilator (MINIVENT mouse ventilator model 845) with the tidal volume adjusted based on body weight (10 µL/g BW). Buprenorphine (0.1 mg/kg BW) was intraperitoneally administered for postoperative pain relief. The left coronary artery was ligated with a 6–0 silk suture just below the left auricular level (*Fraccarollo et al., 2019*; *Galuppo et al., 2017*; *Fraccarollo et al., 2017*; *Fraccarollo et al., 2011*). The suture was passed through a segment of PE-10 tubing. One hour after ischemia the tube was removed to allow for reperfusion. In sham-operated control mice, the ligature around the left anterior descending coronary artery was not tied.

### Isolation of immune cells and fluorescence-activated cell sorting

Mice were anesthetized, intubated and ventilated. Blood samples were drawn from the inferior vena cava into EDTA-containing tubes. Neutrophil count was measured by an automated hematology analyzer (XT 2000i, Sysmex). After lysis of red blood cells with RBC Lysis Buffer (420301; BioLegend), cell suspension was centrifuged at 400 g for 16 min. Single-cell suspensions from heart-draining lymph nodes were prepared with a 70 µm cell strainer and collected by centrifugation at 400 g for 16 min. Mouse femurs were harvested, bone marrows were flushed with FACS-staining buffer, passed through a 70 µm cell strainer (BD Biosciences) and the cell suspension was centrifuged at 400 g for 16 min. The pelleted cells were washed and resuspended in ice-cold FACS-staining buffer. The hearts were perfused for 6 min with the Perfusion Buffer (113 mM NaCl, 4.7 mM KCl, 0.6 mM $KH_2PO_4$, 0.6 mM $Na_2HPO_4$), 1.2 mM MgSO4, 12 mM NaHCO3, 10 mM KHCO3, 10 mM HEPES, 30 mM Taurine, 5.5 mM glucose, 10 mM 2,3-Butanedione monoxime, and subsequently digested for 8 min with the Digestion Buffer (0.2 mg/mL Liberase Roche Diagnostics; and 400 µM calcium chloride in perfusion buffer), using a modified Langendorff perfusion system. The ischemic-reperfused area and surviving myocardium were separated using a dissecting microscope. Subsequently, the heart tissue was smoothly pipetted through a sterile low waste syringe several times in order to obtain a cell suspension in Stop Buffer (perfusion buffer supplemented with 10% (v/v) HI-FCS). The cell suspension was carefully filtered through a 70 µm cell strainer in a 50 mL conical tube, and the cell strainer was washed with perfusion buffer. Then, the cell suspension was centrifuged at 400 g for 20 min. The pelleted cells were washed and resuspended in ice-cold FACS-staining buffer (*Fraccarollo et al., 2019*; *Galuppo et al., 2017*; *Fraccarollo et al., 2017*). To prevent capping of antibodies on the cell surface and non-specific cell labeling all steps were performed on ice and protected from light. Cells were preincubated with Fc Block (Mouse BD Fc Block; BD Biosciences; RRID: AB_394657) for 10 min. Subsequently, fluorochrome-conjugated antibodies were added and incubated for 30 min. Finally, the cells were washed twice with ice-cold FACS-staining buffer. After preselection in side scatter (SSC) vs. forward scatter (FSC) dot plot to exclude debris and FSC vs. Time-of-Flight (ToF) dot plot to discriminate doublets by gating single cells, blood monocytes were identified as CD45$^+$/CD11b$^+$/Ly6G$^-$/CD115$^+$ cells, blood neutrophils as CD45$^+$/CD11b$^+$/Ly6G$^+$ cells, infarct macrophages as CD45$^+$/CD11b$^+$/Ly6G$^-$/F4/80$^+$ cells and infarct neutrophils as CD45$^+$/CD11b$^+$/F4/80$^-$/Ly6G$^+$ cells (*Fraccarollo et al., 2019*; *Galuppo et al., 2017*; *Fraccarollo et al.,*

*2017*). The following antibodies were used: anti-CD45 (clone 104, 1:100, BioLegend; RRID:AB_893350; clone 30-F11, 1:100, BD Biosciences; RRID:AB_394003); anti-F4/80 (clone BM8, 1:100, BioLegend, RRID:AB_2734779; clone T45-2342, 1:100, BD Biosciences; RRID:AB_893481); anti-CD11b (clone M1/70, 1:100, eBioscience, RRID:AB_2637408; BD Biosciences, RRID:AB_394002); anti-CD4 (clone GK1.5, 1:100, BioLegend; RRID:AB_389302); anti-CD115 (clone AFS98, 1:100, BioLegend; RRID:AB_2562760, RRID:AB_11218983); anti-Ly6G (clone 1A8, 1:100, BioLegend, RRID:AB_1877163; 1:200, BD Biosciences, RRID:AB_394208, RRID:AB_2739207); anti-CXCR2 (clone SA045E1, 1:100, BioLegend; RRID:AB_2565563, RRID:AB_2565689); anti-CD101 (clone Moushi101, 1:100, eBioscience, RRID:AB_1210729; clone 307707, 1:100, BD Biosciences, RRID:AB_2738821). For MMP-9 and IL-1ß intracellular staining, the Cytofix/Cytoperm Fixation/Permeabilization Kit was used according to the manufacturer's protocol (BD Biosciences). Antibodies included anti-MMP-9 (AF909, 1:100, R and D Systems; RRID:AB_355706); anti IL-1ß (ab9722, 1:100, Abcam; RRID:AB_308765); donkey anti-goat secondary antibody (A-11055, Invitrogen; RRID:AB_2534102) and goat anti-rabbit secondary antibody (A-11034, Invitrogen; RRID:AB_2576217). FMO controls were included during acquisition for gating analyses to distinguish positive from negative staining cell populations. FACS data were acquired on a Gallios flow cytometer and analyzed with Gallios software (Beckman Coulter). Cell sorting was performed using a FACS Aria Fusion (BD Biosciences). Cells were sorted in Lysis-Buffer (PrepEase RNA Spin Kit, Affymetrix; RNeasy Plus Mini Kit; QIAGEN), (*Fraccarollo et al., 2019*; *Galuppo et al., 2017*) or in sterile Sorting Medium. For morphological analysis, CD101$^{neg}$/CD101$^{pos}$ neutrophils and mediastinal lymph node cell suspensions were centrifuged onto cytospin slides and stained with May-Grünwald Giemsa or Wright-Giemsa. For assessment of apoptosis FACS-sorted CD101$^{neg}$/CD101$^{pos}$ neutrophils were cultured for 24 hr in RPMI 1640 medium supplemented with 5% HI-FCS and 1% PenStrep. Apoptotic rate was determined by flow cytometry using Vybrant DyeCycle Violet stain.

## RNA-Seq

Total RNA was isolated using PrepEase RNA Spin Kit according to the manufacturer's instructions (*Fraccarollo et al., 2019*; *Galuppo et al., 2017*). Sorted cells were directly collected in lysis buffer and immediately processed. RNA quantification and quality testing were assessed by NanoDrop 2000 (Thermo Fisher Scientific) and Bioanalyzer 2100 (Agilent). Libraries for RNA sequencing were prepared from 30 ng total RNA; from each sample, polyA RNA was purified, converted to cDNA and linked to Illumina adapters using the Illumina TruSeq stranded mRNA Kit according to the manufacturer's instructions. Samples were multiplexed and sequenced on an Illumina NextSeq 500 in a 75 nt single end setting using a high-output run mode. Raw BCL files were demultiplexed and converted to sample-specific FASTQ files using bcl2fastq v1.8.4 (Illumina). Residual adapter sequences present in the sequencing reads were removed with Cutadapt version 1.12. Reads (~ 40 million per sample) were aligned to the mouse reference sequence GENCODE vM8 using STAR version 2.5.2b. RNA sequencing data analysis was undertaken with the statistical programming language, R. The R package DeSeq2 (v1.14.1) was used to evaluate differential gene expression (*Fraccarollo et al., 2019*; *Galuppo et al., 2017*).

## Statistical analysis

Data are presented as mean ± SEM or as median [interquartile range] as indicated. Normality of data was assessed by Shapiro-Wilk test. Normal data were analyzed by one-way ANOVA with Tukey post hoc test, otherwise by Kruskal-Wallis test for comparisons of median values, with Mann-Whitney *U* test for multiple comparisons. Unpaired t-test or Mann-Whitney *U* test was used to compare two independent groups. Linear regression analysis or Spearman's rank correlation test was used to determine relationship between variables. Values of $p \leq 0.05$ were considered statistically significant.

## Results

### Increased circulating levels of CD14$^{+}$HLA-DR$^{neg/low}$ monocytes in patients with acute MI

Flow cytometric immunophenotyping was performed in whole blood from patients with unstable angina (UA) or acute MI within 24–72 hr of symptom onset (median 43.6 hr). A time course analysis

of monocyte subset-frequencies and circulating levels up to day five after MI is shown in *Figure 1—figure supplement 1*.

NSTEMI/STEMI patients displayed significantly higher absolute neutrophil and monocyte counts versus UA patients (*Table 2*). Based on HLA-DR/CD14/CD16 expression, monocytes can be divided into different subsets. We detected increased circulating levels of intermediate (HLA-DR$^{++}$CD14$^{++}$CD16$^+$CX3CR1$^+$) in ACS patients versus control, and of non-classical (HLA-DR$^+$CD14$^+$CD16$^{++}$CX3CR1$^{++}$) in STEMI versus UA patients and controls (*Table 2*, *Figure 1—figure supplement 2*). There were no significant correlations between intermediate/non-classical monocytes and LV ejection fraction/CK$_{max}$.

We found increased percentages and absolute numbers of circulating CD14$^+$HLA-DR$^{neg/low}$ monocytes in STEMI/NSTEMI patients as compared to UA patients (*Figure 1A and B* and *Figure 1—figure supplement 2B*). Linear regression analysis revealed a positive correlation between circulating levels of CD14$^+$HLA-DR$^{neg/low}$ monocytes and CK$_{max}$ (*Figure 1C*) and a negative correlation with LV ejection fraction (R=0.44, p<0.001).

Receiver operating characteristic (ROC) curve analysis based on circulating CD14$^+$HLA-DR$^{neg/low}$ monocytes (n/µL), discriminating UA and STEMI patients revealed an AUC of 0.949 (95% CI: 0.892–1; p<0.001), whereas a lower AUC discriminating UA and NSTEMI patients was observed (AUC=0.786; 95% CI: 0.612–0.961; p<0.01). By combining CD14$^+$HLA-DR$^{neg/low}$ monocytes with CK$_{max}$ AUC was increased to 0.970; (95% CI: 0.931–1) (*Figure 1D*) but not in combination with LVEF (AUC=0.925; 95% CI: 0.840–1) compared to CD14$^+$HLA-DR$^{neg/low}$ monocytes alone discriminating UA and STEMI patients.

Next, we analyzed the immunoregulatory features of CD14$^+$HLA-DR$^{neg/low}$ monocytes. Using FACS-sorting, CD14$^+$HLA-DR$^{neg/low}$/CD14$^+$HLA-DR$^{high}$ cells were isolated from blood of patients with AMI (*Figure 2A*). Quantitative RT-PCR showed that HLA-DR$^{neg/low}$ monocytes express high amounts of S100A9 and IL1R1 (*Figure 2B*). Of interest, studies in heart failure patients have provided evidence for the presence of HLA-DR$^{neg/low}$ cells within myocardial tissue expressing high levels of S100A9 (*Bajpai et al., 2018*).

CD14$^+$HLA-DR$^{neg/low}$ monocytes did not suppress T-cell proliferation (*Figure 2C*), indicating that the expanded population of monocytic cells in infarct patients are not immunosuppressive. Remarkably, macrophages differentiated from CD14$^+$HLA-DR$^{neg/low}$ monocytes by 4-day culture with M-CSF produced more TNF-α, IL-6, and IL-1ß upon stimulation with IFN-γ, as compared to macrophages generated from monocytes CD14$^+$HLA-DR$^{high}$ (*Figure 2D–F*).

These results may indicate an important role for CD14$^+$HLA-DR$^{neg/low}$ monocytes in the inflammatory response during AMI.

No difference was seen in the expression of CAT, CCR1, IL1R2, LCN2, MMP8, NOS2, SAAP3, and STAT3 (*Figure 2—figure supplement 1A*) factors dysregulated in circulating monocytes as well as in infarct macrophages in a mouse model of reperfused AMI (*Figure 2—figure supplement 1B and C*).

**Table 2.** Leukocyte count and monocyte subsets.

| | CTR (*N* = 17) | UA (*N*=11) | NSTEMI (*N*=16) | STEMI (*N*=44) | p (K-W) |
|---|---|---|---|---|---|
| Neutrophil (10$^3$/µL) | 3.25 (2.74–3.42) | 4.05 (3.65–4.56)* | 5.72 (4.80–7.79)*† | 6.13 (5.18–7.04)*† | <0.0001 |
| Monocyte (10$^3$/µL) | 0.65 (0.53–0.74) | 0.74 (0.49–0.80) | 0.88 (0.72–1.03)*† | 0.99 (0.77–1.25)*† | <0.001 |
| Lymphocyte (10$^3$/µL) | 2.20 (1.96–2.47) | 1.82 (1.55–1.98) | 1.97 (1.77–2.49) | 2.07 (1.58–2.64) | 0.30 |
| Lymphocyte/neutrophil ratio | 0.70 (0.57–0.79) | 0.45 (0.38–0.56)* | 0.32 (0.28–0.44)* | 0.35 (0.27–0.45)* | <0.0001 |
| Eosinophil (10$^3$/µL) | 0.16 (0.10–0.30) | 0.15 (0.12–0.16) | 0.21 (0.14–0.34) | 0.12 (0.06–0.20) | 0.08 |
| Classical monocyte (*n*/µL) | 476 (334–583) | 332 (244–388) | 510 (454–719)† | 505 (388–666)† | <0.05 |
| Intermediate monocyte (*n*/µL) | 130 (73–145) | 186 (131–367)* | 204 (144–310)* | 249 (167–442)* | <0.001 |
| Non-classical monocyte (*n*/µL) | 48 (30–64) | 50 (37–67) | 64 (47–108) | 102 (60–138)*† | <0.001 |

Data are presented as median (IQR). Kruskal-Wallis (K-W) test; *p<0.05 vs. Control (CTR); †p<0.05 vs. UA.

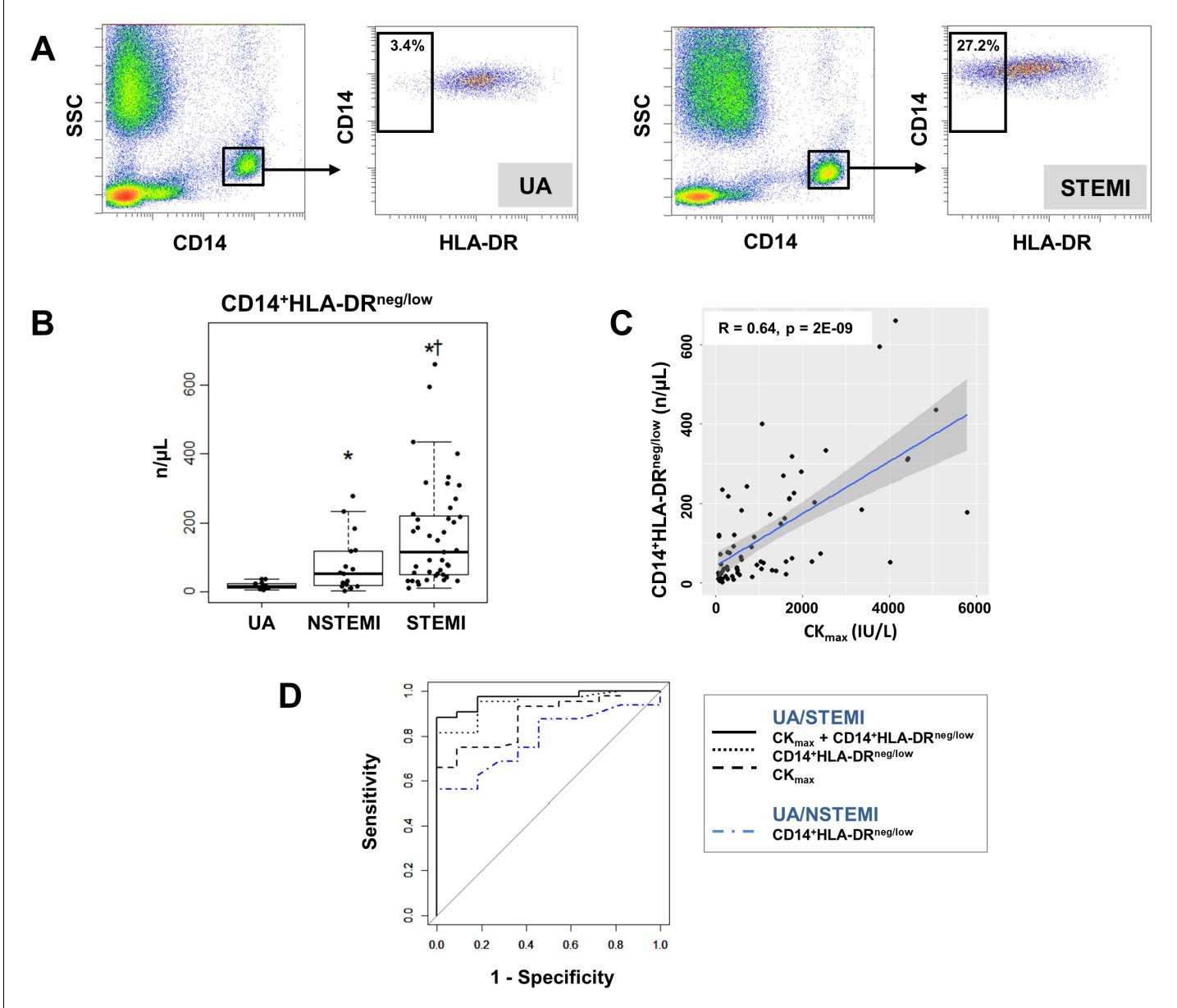

**Figure 1.** Increased circulating levels of CD14+HLA-DRneg/low monocytes in patients with AMI. (A) Gating strategy to identify CD14+HLA-DRneg/low monocytes. (B) Circulating levels of CD14+HLA-DRneg/low monocytes in patients with unstable angina (UA; n=11), non-ST-elevation MI (NSTEMI, n=16), and ST-elevation MI (STEMI, n=44). (C) Linear regression analysis between circulating levels of CD14+HLA-DRneg/low monocytes and maximum CK ($CK_{max}$) in patients with acute coronary syndrome. (D) Receiver operator characteristic (ROC) curve of CD14+HLA-DRneg/low monocytes discriminating UA/STEMI and UA/NSTEMI patients and the combination of CD14+HLA-DRneg/low monocytes (n/µL) with $CK_{max}$. *$p<0.05$, vs. UA; †$p<0.05$, vs. NSTEMI. The online version of this article includes the following figure supplement(s) for figure 1:

**Figure supplement 1.** Time course analysis of subset-frequencies and circulating levels of monocytes in patients with unstable angina (UA; n=11), non-ST-elevation MI (NSTEMI, n=16), and ST-elevation MI (STEMI, n=44).

**Figure supplement 2.** Monocyte subset-frequencies.

## Immature CD10neg neutrophils expand in the peripheral blood from patients with acute MI

Phenotypic characterization of neutrophils was performed in whole blood. The absolute numbers and frequency of circulating CD16+CD66b+CD10neg neutrophils were significantly increased in STEMI versus UA patients (*Figure 3A and B* and *Figure 3—figure supplement 1A*). A time course

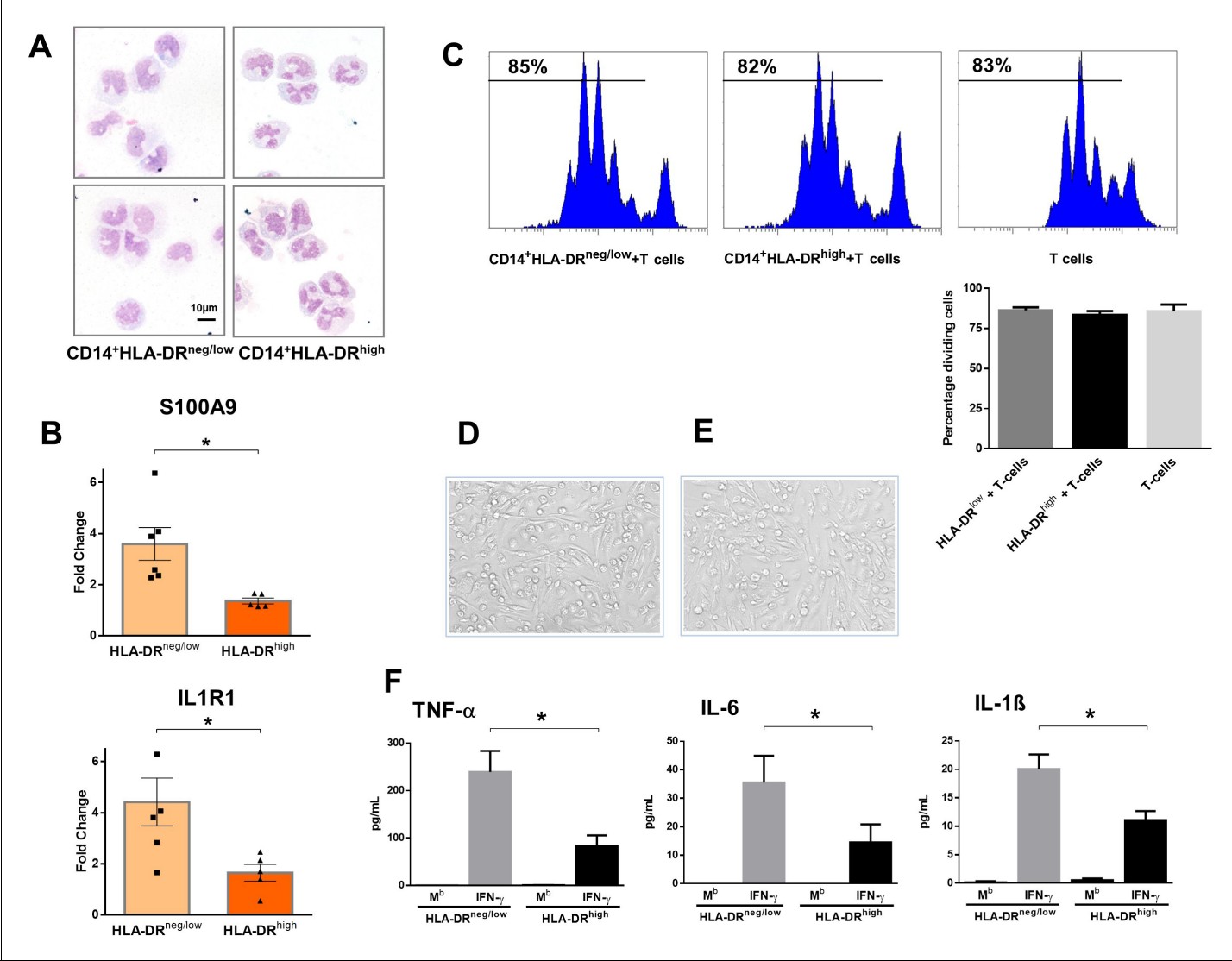

**Figure 2.** CD14+HLA-DR^neg/low monocytes from patients with AMI are not immunosuppressive but exhibit an inflammatory phenotype. (A) May-Grünwald Giemsa stained cytospin preparations of CD14+HLA-DR^neg/low and CD14+HLA-DR^high monocytes. (B) Relative RNA expression of S100A9 and IL1R1 in CD14+HLA-DR^neg/low versus CD14+HLA-DR^high monocytes. (C) T-cell proliferation in the presence of CD14+HLA-DR^neg/low or CD14+HLA-DR^high monocytes assessed by CellTrace Violet dilution after 96 hr of co-culture. (D) Macrophages differentiated from CD14+HLA-DR^neg/low monocytes and (E) CD14+HLA-DR^high cells by 4-day culture with M-CSF. (F) TNF-$\alpha$, IL-6, and IL-1ß in supernatants of macrophage cultures upon stimulation with IFN-$\gamma$. M^b=baseline. CD14+HLA-DR^neg/low/CD14+HLA-DR^high cells were isolated by flow-cytometric sorting from patients with AMI (n=5–6). Data are presented as mean ± SEM. *p<0.05.

The online version of this article includes the following figure supplement(s) for figure 2:

**Figure supplement 1.** Expression of genes dysregulated in circulating monocytes and infarct macrophages in a mouse model of reperfused AMI.

analysis of circulating levels and frequencies of CD16+CD66b+CD10^neg neutrophils up to day 5 after MI is shown in *Figure 3—figure supplement 1B*. Circulating levels of CD16+CD66b+CD10^neg neutrophils correlated positively with $CK_{max}$ (*Figure 3C*) and negatively with LV ejection fraction (R=0.4, p<0.001).

ROC curve analysis of circulating CD10^neg neutrophils (n/µL), discriminating UA and STEMI patients revealed an AUC of 0.798 (95% CI: 0.683–0.913; p<0.001) but a lower AUC discriminating UA and NSTEMI patients (AUC=0.687; 95% CI: 0.482–0.892; p=0.015). By combining CD10^neg neutrophils with $CK_{max}$ or LVEF AUC was increased to 0.909; (95% CI: 0.831–0.986) and to 0.833 (95% CI: 0.691–0.974), respectively discriminating UA and STEMI patients (*Figure 3D*). Of note, AUC

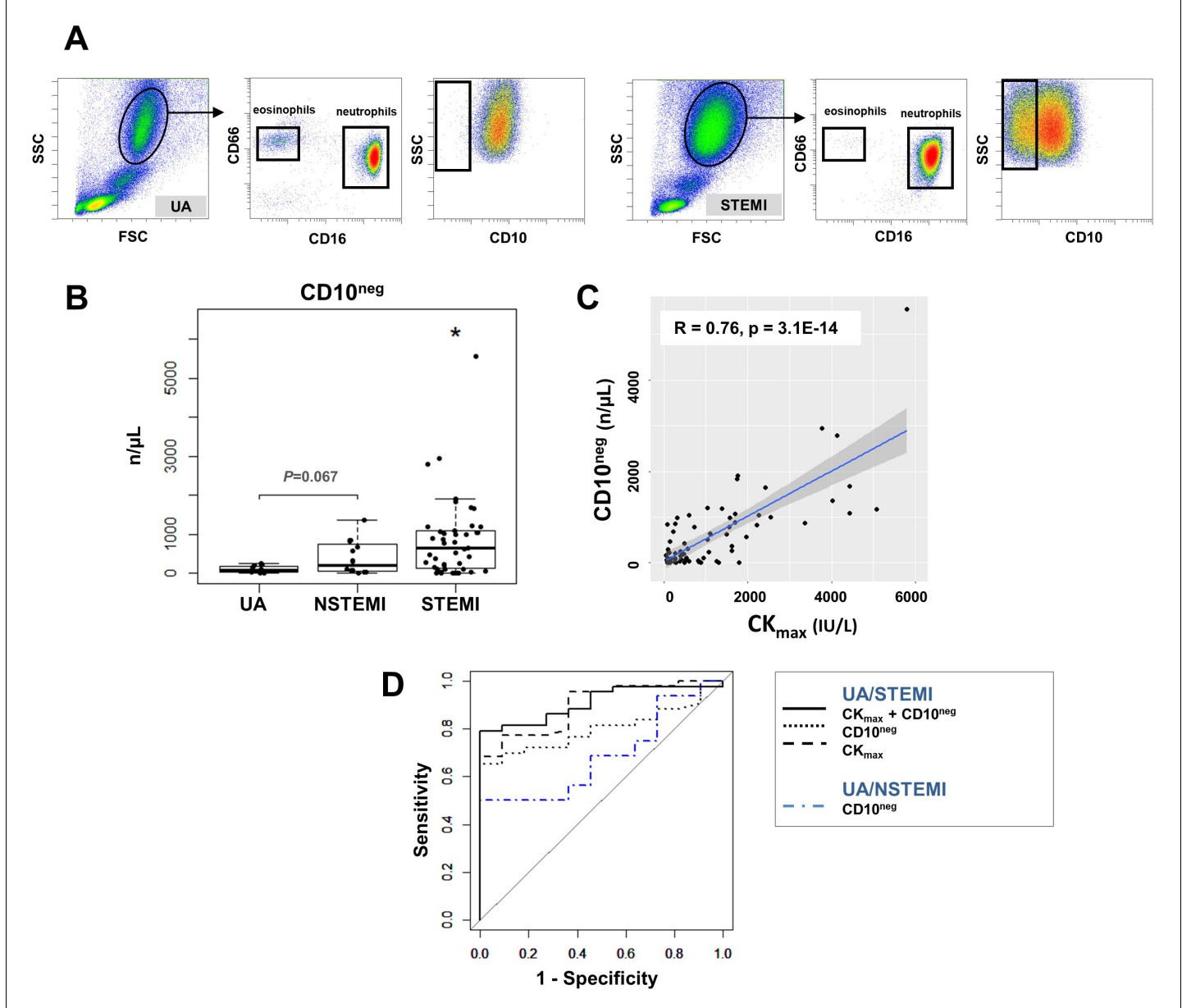

**Figure 3.** Circulating normal-density CD10[neg] neutrophils increase in patients with AMI. (A) Gating strategy to identify CD10[neg] neutrophils. (B) Circulating levels of $CD16^+CD66b^+CD10^{neg}$ neutrophils in patients with unstable angina (UA; n=11), non-ST-elevation MI (NSTEMI, n=16), and ST-elevation MI (STEMI, n=44). (C) Linear regression analysis between circulating levels of CD10[neg] neutrophils and maximum CK ($CK_{max}$). (D) Receiver operator characteristic (ROC) curve of CD10[neg] neutrophils (n/μL) discriminating UA/STEMI and UA/NSTEMI patients and the combination of CD10[neg] neutrophils with $CK_{max}$ in patients with acute coronary syndrome. *p<0.05 vs. UA.

The online version of this article includes the following figure supplement(s) for figure 3:

**Figure supplement 1.** Increased frequency of CD10[neg]neutrophils in patients with AMI.

derived from ROC curve analysis combining HLA-DR[neg/low] monocytes and CD10[neg] neutrophils, discriminating UA/STEMI patients, resulted equal to 0.970 with 95% CI: 0.927–1, higher compared to AUC value obtained by each parameter individually.

$CD16^+CD66b^+CD10^{neg}$ neutrophils co-purified with the erythrocyte fraction following density gradient centrifugation. Low-density neutrophils were not present in mononuclear cell fraction obtained from AMI patients. Recently, the work of *Maréchal et al., 2020* found higher frequency of

immature low-density neutrophils in STEMI patients. However, cancer patients were not excluded which might have influenced the proportion of low-density CD10$^{neg}$ cells.

Cytospin slides were made after FACS-sorting to examine nuclear morphology (*Figure 4A*). We found that the majority of the CD16$^{+}$CD66b$^{+}$CD10$^{neg}$ cells has an immature morphology with a lobular nucleus, while CD16$^{+}$CD66b$^{+}$CD10$^{pos}$ cells are mature neutrophils with segmented nuclei (*Figure 4A*). These findings were obtained when neutrophils were isolated by dextran sedimentation as well as by negative selection using magnetic beads, indicating that the differences between the

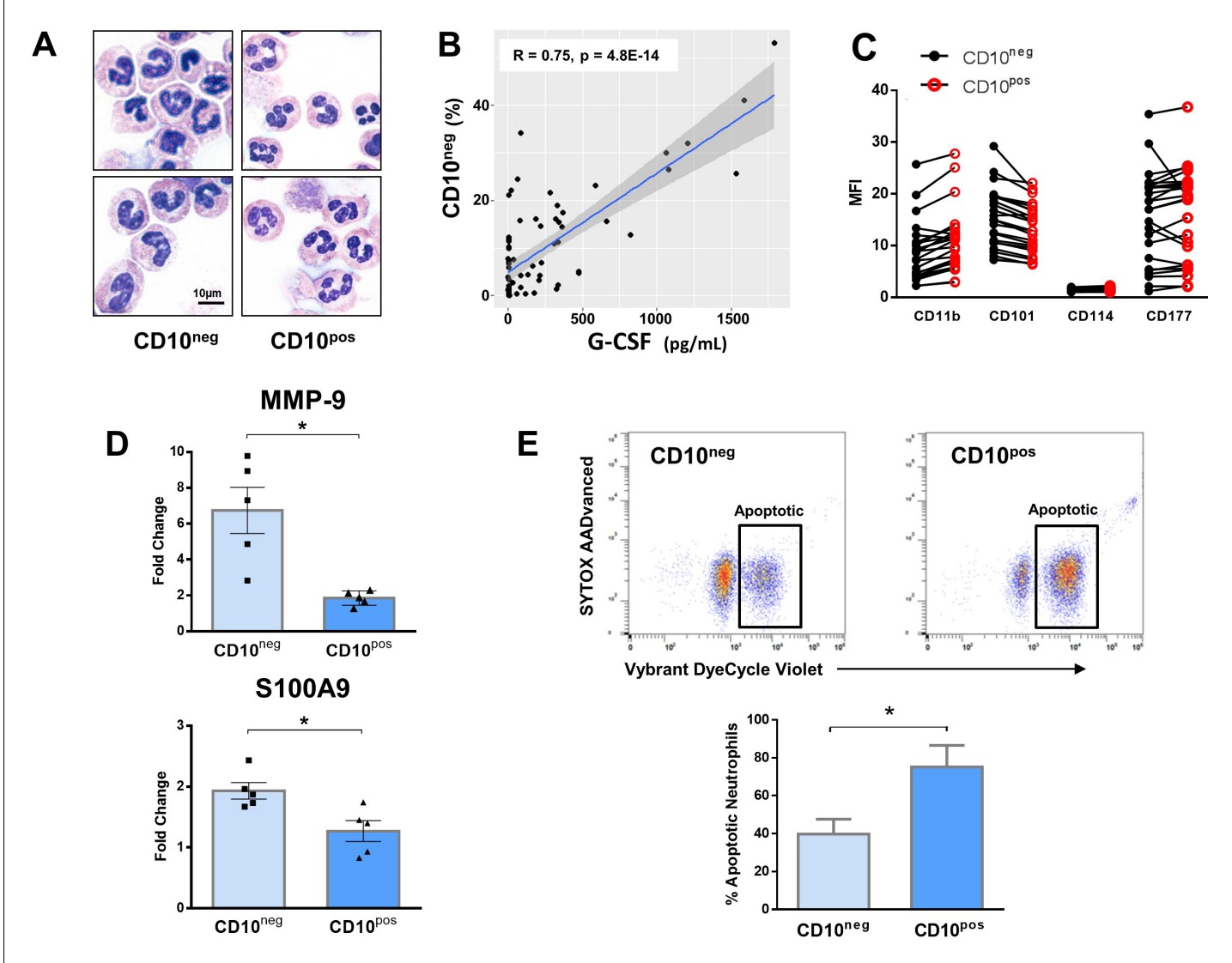

**Figure 4.** Immature CD10$^{neg}$ neutrophils from patients with AMI express high amounts of MMP-9 and S100A9 and display resistance to apoptosis. (A) May-Grünwald Giemsa stained cytospin preparations of CD16$^{+}$CD66b$^{+}$CD10$^{neg}$ (CD10$^{neg}$) and CD16$^{+}$CD66b$^{+}$CD10$^{pos}$ (CD10$^{pos}$) neutrophils. (B) Linear regression analysis between the percentages of CD16$^{+}$CD66b$^{+}$CD10$^{neg}$ neutrophils and circulating levels of G-CSF in patients with acute coronary syndrome (n=71). (C) Mean fluorescence intensity (MFI) of CD11b, CD101, CD114, and CD177 on CD10$^{neg}$ versus CD10$^{pos}$ neutrophils (n=25). (D) Relative RNA expression of MMP-9 and S100A9 in CD10$^{neg}$ versus CD10$^{pos}$ neutrophils. (E) Percentage of apoptotic neutrophils assessed by flow cytometry using Vybrant DyeCycle Violet stain and SYTOX AADvanced stain. CD10$^{neg}$/CD10$^{pos}$ neutrophils were isolated by flow-cytometric sorting from patients with AMI (n=4–5). *p<0.05.

The online version of this article includes the following figure supplement(s) for figure 4:

**Figure supplement 1.** Expression of genes regulated in circulating neutrophils and infarct neutrophils in a mouse model of reperfused AMI.
**Figure supplement 2.** Immature CD101$^{neg}$ neutrophils are rapidly released into the bloodstream after reperfusion in a mouse model of AMI.
**Figure supplement 3.** Interactions between CD10$^{neg}$neutrophils and CD14$^{+}$HLA-DR$^{neg/low}$monocytes.

neutrophil subpopulations cannot be considered an artifact due to the isolation technique used (*Hardisty et al., 2021*). Of note, linear regression analysis revealed a strong positive correlation between the percentages of CD16[+]CD66b[+]CD10[neg] cells and circulating levels of G-CSF (*Figure 4B*). AMI patients with higher systemic concentrations of G-CSF have increased CD10[neg] neutrophils levels, suggesting G-CSF-driven immature neutrophil release/expansion.

## Immature CD10[neg] neutrophils from patients with AMI express high amounts of MMP-9 and S100A9 and display resistance to apoptosis

In an attempt to identify-specific surface antigens that characterize normal-density CD10[neg] neutrophils derived from AMI patients, we next examined the expression of markers associated with neutrophil maturation/activation (*Hardisty et al., 2021*; *Silvestre-Roig et al., 2019*). Flow cytometry showed that CD11b, CD101, CD114, and CD177 were expressed at equivalent levels on CD10[neg] neutrophils versus CD10[pos] neutrophils (*Figure 4C*).

Next, we investigated the expression of inflammatory mediators playing a pathophysiological role in wound healing after AMI. CD10[neg] neutrophils sorted from blood of AMI patients express higher amounts of MMP-9 and S100A9 than CD10[pos] neutrophils (*Figure 4D*). No difference was found in the expression of IRG1 (ACOD1), IL1R1, MMP8, NOS2, and STAT3 (*Figure 4—figure supplement 1A*), genes regulated in circulating neutrophils as well as in infarct neutrophils in a mouse model of reperfused AMI (*Figure 4—figure supplement 1A–1C*).

Mouse neutrophils, unlike human granulocytes, lack CD10 expression (*Kalled et al., 1995*) Using next-generation RNA sequencing we identified CD101 among the genes down-regulated by ischemia in circulating neutrophils (*Figure 4—figure supplement 1B*).

Next, we found that CD101 can be used as a surface marker to identify the mature neutrophil subset among the heterogeneous Ly6G[pos]CXCR2[pos] neutrophil populations, released into the bloodstream after ischemia/reperfusion (*Figure 4—figure supplement 2A and B*). As revealed by morphological analysis circulating CD11b[bright]CD101[pos] neutrophils have a mature morphology, whereas CD11b[dim]CD101[neg] cells are immature neutrophils with ring-shaped nuclei (*Figure 4—figure supplement 2B*). Recent study in mice showed that CD101 segregates immature neutrophils from mature neutrophils during G-CSF stimulation and in the tumor setting (*Evrard et al., 2018*). Notably, the levels of CD101[neg] neutrophils in the bone marrow of mice subjected to ischemia followed by 6 hr of reperfusion were markedly reduced (*Figure 4—figure supplement 2C*), indicating early release of immature neutrophils from bone marrow into the circulation after AMI. Moreover, we found that CD11b[pos]Ly6G[pos]CD101[neg] cells sorted from peripheral blood after AMI express higher amounts of Irg1 (Acod1), Il1r1, Mmp8, Nos2, and Stat3 than CD101[pos] neutrophils and showed markedly reduced CD101 expression (*Figure 4—figure supplement 1D and E*). Taken together, our results emphasize interspecies differences in immune responses after ischemic injury, suggesting caution in applying findings in murine models of myocardial ischemia/reperfusion to patients with AMI.

To further define the functional properties of the immature neutrophil populations derived from AMI patients we investigated spontaneous apoptosis. The apoptotic rate of CD10[neg] neutrophils was substantially reduced compared with CD10[pos] neutrophils after 24 hr in vitro culture as determined by flow cytometric analysis using Vybrant DyeCycle Violet/SYTOX AADvanced stain (*Figure 4E*) as well as Apotracker Green (*Figure 4—figure supplement 3A*). The decreased apoptosis rate of CD10[neg] neutrophils was reflected by an increase in the percentage of live cells. Morphological assessment of cytospins (*Figure 4—figure supplement 3B*) confirmed the results obtained by flow cytometry.

Neutrophils may play a crucial role in cardiac healing after AMI by influencing macrophage polarization (*Horckmans et al., 2017*). *Horckmans et al., 2017* found an induction of MerTK expression in macrophages (differentiated from peripheral blood monocytes of healthy donors) exposed to IL-4 in the presence of neutrophil supernatants from healthy individuals. Therefore, we next assessed MerTK expression on macrophages differentiated from CD14[+]HLA-DR[neg/low] and CD14[+]HLA-DR[high] monocytes and polarized in the presence of supernatants derived from CD10[neg] neutrophils as well as CD10[pos] neutrophils (*Figure 4—figure supplement 3C and D*). CD14[+]HLA-DR[neg/low]/CD14[+]HLA-DR[high] cells and CD10[neg]/CD10[pos] neutrophils were isolated from patients with AMI. We found a significantly higher expression of MerTK on CD14[pos]CD163[pos] macrophages differentiated from CD14[+]HLA-DR[neg/low] monocytes and stimulated with IFN-γ and IL-4 in presence of supernatants from CD10[pos] neutrophils (*Figure 4—figure supplement 3C*). Efficient efferocytosis by human

macrophages requires MerTK induction (*Zizzo et al., 2012*). Thus, we speculated that factors secreted by CD10$^{pos}$ neutrophils from patients with AMI may enhance the capacity of HLA-DR$^{neg/low}$ monocyte-derived macrophages to phagocyte apoptotic cells by regulating MerTK expression.

The alarmin S100A9 have been shown to have chemotactic activity for monocytes (*Chen et al., 2015*). Previous experiments have focused on monocytic THP-1 cells and monocytes isolated from healthy human subjects (*Chen et al., 2015*). Here, we analyzed S100A9-induced chemotaxis of CD14$^+$HLA-DR$^{neg/low}$/CD14$^+$HLA-DR$^{high}$ monocytes from patients with AMI. As shown in *Figure 4—figure supplement 3E*, S100A9 significantly induced the migration of CD14$^+$HLA-DR$^{high}$ as well as CD14$^+$HLA-DR$^{neg/low}$ monocytes. Therefore, it is also possible that S100A9 regulates CD14$^+$HLA-DR$^{neg/low}$/CD14$^+$HLA-DR$^{high}$ monocytes recruitment during AMI by functioning as a chemoattractant.

## CD10$^{neg}$ neutrophils and HLA-DR$^{neg/low}$ monocytes are linked to levels of immune-inflammation markers

In a subgroup of patients we measured serum levels of immune inflammation markers (*Table 3*). MMP-9, S100A9/S100A8, NGAL, IL-6, and IL-1ß levels were higher in STEMI patients versus UA patients.

Spearman's correlation matrix showed multiple intercorrelations among CD10$^{neg}$ neutrophils, HLA-DR$^{neg/low}$ monocytes and several inflammation markers (*Figure 5*). The percentages of CD14$^+$-HLA-DR$^{neg/low}$ cells significantly correlated with circulating levels of MMP-9, S100A9/S100A8, IL-6, IL-1ß, TNF-α, MPO, and NGAL. Noticeable, CD10$^{neg}$ neutrophils, which expand proportional to the degree of myocardial injury, significantly correlated with levels of MMP-9, S100A9/S100A8, NGAL, MPO, IL-6, and IL-1ß (*Figure 5*).

## Immature CD101$^{neg}$ neutrophils are recruited to sites of cardiac injury and exhibit resistance to apoptosis

We then investigated whether immature CD101$^{neg}$ neutrophils are recruited to the injured myocardium using the mouse model of reperfused AMI. Preliminary experiments showed that current protocols for tissue dissociation and the recovery of neutrophils from ischemic myocardium involving long enzymatic digestion times resulted in cell activation/damage and phenotypic changes that can be easily over-interpreted. Therefore, using a modified Langendorff perfusion system, the infarcted hearts were perfused for 6 min to remove blood cells and subsequently digested for only 8 min to preserve cell surface antigens along with expression profiles. Flow cytometry analysis of immune cells isolated from the ischemic region 3 hr after reperfusion revealed marked infiltration of CD101$^{neg}$ neutrophils, displaying increased expression of the matrix-degrading protease MMP-9 (*Figure 6A*). Moreover, as shown in *Figure 6B*, we found that 24 hr after reperfusion CD101$^{neg}$ neutrophils expressed IL-1ß at higher levels compared to CD101$^{pos}$ cells. Morphological analysis of cytospin preparations of sorted neutrophils showed that CD101$^{neg}$ neutrophils isolated from ischemic myocardium 24 hr after reperfusion still have an immature morphology (*Figure 6C*). These findings suggest migration and homing of immature CD101$^{neg}$ neutrophils to ischemic myocardium shortly after reperfusion.

**Table 3.** Immune inflammation markers.

| | UA (*N*=11) | NSTEMI (*N*=10) | STEMI (*N*=26) | p (K-W) |
|---|---|---|---|---|
| MMP-9 (ng/mL) | 429 (320-461) | 447 (324-597) | 544 (466-758)* | <0.01 |
| S100A8/A9 (ng/mL) | 7332 (4638–9461) | 13802 (9152–21066)* | 17352 (8592–27830)* | <0.05 |
| NGAL (ng/mL) | 264 (198-318) | 328 (211-473) | 417 (312-653)* | <0.05 |
| MPO (ng/mL) | 221 (153-337) | 323 (158-443) | 389 (230-487)* | 0.05 |
| IL-6 (pg/mL) | 11.2 (9.2–21.1) | 30.6 (24.5–57.4)* | 47.7 (22.0–102.1)* | <0.01 |
| TNF-α (pg/mL) | 1.8 (1.3–15.7) | 4.6 (2.9–7.2) | 12.1 (5.0–21.8) | 0.14 |
| IL-1ß (pg/mL) | 2.4 (2.3–2.9) | 4.2 (2.4–7.9) | 10.0 (2.5–16.4)* | 0.05 |

Data are presented as median (IQR). Kruskal-Wallis (K-W) test; *p<0.05 vs. UA.

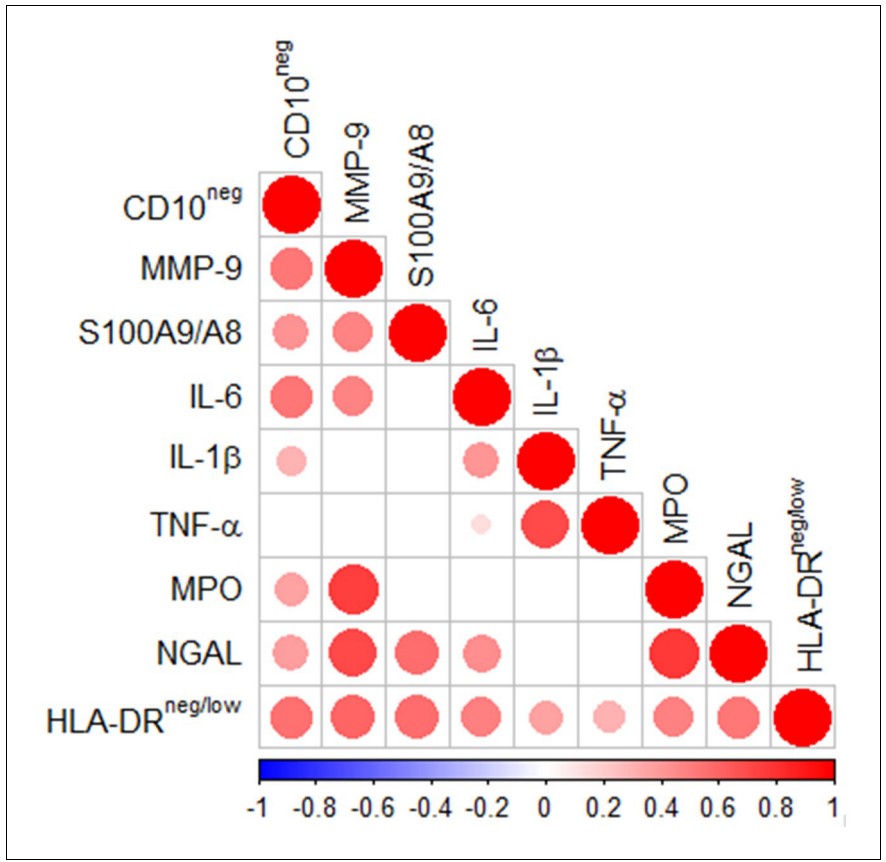

**Figure 5.** Multiple intercorrelations among CD10[neg] neutrophils, HLA-DR[neg/low] monocytes and immune-inflammation markers. Spearman-correlation matrix of CD16[+]CD66b[+]CD10[neg] neutrophils (%), CD14[+]HLA-DR[neg/low] monocytes (%) and circulating levels of MMP-9, S100A9/S100A8, IL-6, IL-1ß, TNF-α, MPO, and NGAL (levels). UA (n=11), NSTEMI (n=10), and STEMI (n=26). Each circle illustrates a significant correlation between different couples of parameters (p<0.05). The correlation coefficient is colored and sized up according to the value; square leaved blank indicates not significant correlation.

Effective resolution of inflammation as well as a favorable wound healing response requires timely phagocytic clearance of apoptotic neutrophils by macrophages. As CD10[neg] neutrophils derived from AMI patients showed reduced spontaneous apoptosis, we studied the apoptosis rate of CD101[neg] and CD101[pos] neutrophils isolated from the ischemic region 24 hr after reperfusion. We found that CD101[neg] neutrophils were more resistant to apoptosis than their CD101[pos] counterparts. After 24 hr in vitro culture the majority of CD101[neg] neutrophils remained alive, as assessed by flow cytometry (*Figure 6D*).

## Immature CD101[neg] neutrophils are present in mediastinal lymph nodes after acute MI

A crucial role for neutrophils in the orchestration of adaptive immunity is emerging (*Costa et al., 2019*; *Silvestre-Roig et al., 2020*). During homeostasis and inflammation, neutrophils can migrate to lymph nodes, which are critical sites for T-cell priming (*Voisin and Nourshargh, 2019*). Therefore, we next addressed whether immature CD101[neg] neutrophils migrate into heart-draining lymph nodes early after AMI. We found a small population of neutrophils in heart-draining lymph nodes at steady state, consistent with recent studies (*Lok et al., 2019*). Notably, in mice subjected to coronary occlusion followed by 3 hr of reperfusion the percentage of neutrophils in mediastinal lymph nodes increased significantly (*Figure 7A*). About 50% of infiltrating neutrophils were CD101[neg] cells. CD101[neg] neutrophils were nearly absent in mediastinal lymph nodes at steady state (*Figure 7A*). Analysis of cytospins of mediastinal lymph node cell suspensions revealed the presence of neutrophils with ring-shaped nuclei as well as mature neutrophils with segmented nuclei after AMI

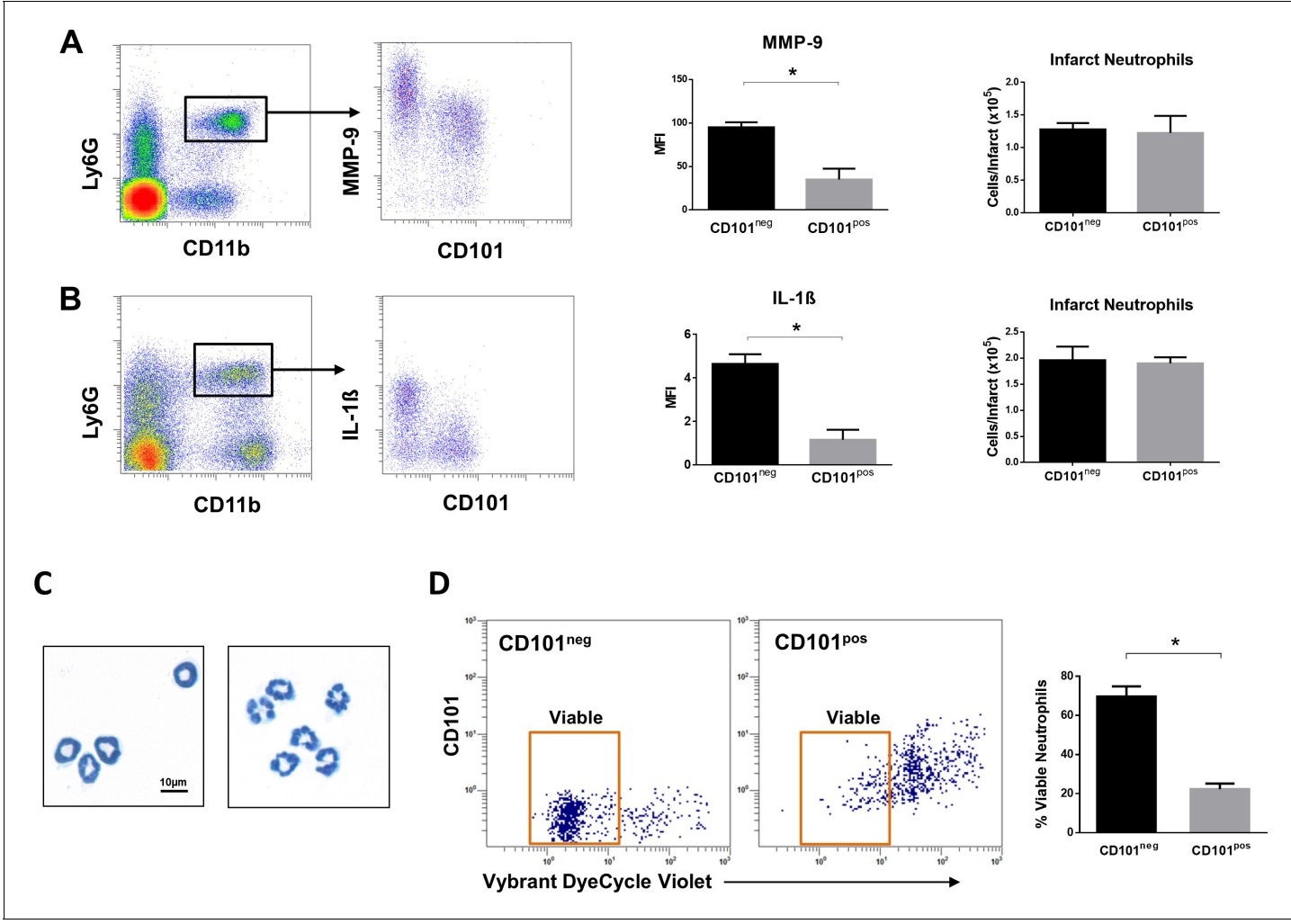

**Figure 6.** Immature CD101[neg] neutrophils are rapidly recruited to ischemic sites, are a major source of MMP-9 and IL-1ß in the reperfused myocardium and exhibit resistance to apoptosis. (A) Flow cytometric gating strategy to identify neutrophils in the ischemic region, mean fluorescent intensity (MFI) of MMP-9 on CD101[neg] and CD101[pos] neutrophils, number of CD101[neg] and CD101[pos] neutrophils 3 hr after reperfusion. (B) Flow cytometry identifying infarct neutrophils, mean fluorescent intensity of IL-1ß on CD101[neg] and CD101[pos] neutrophils, number of CD101[neg] and CD101[pos] neutrophils 24 hr after reperfusion. (C) Wright-Giemsa stained cytospin preparations of CD101[neg] and CD101[pos] neutrophils isolated by cell sorting from ischemic myocardium 24 hr after reperfusion. (D) Rate of apoptosis of infarct CD101[neg]/CD101[pos] neutrophils determined by flow cytometry using Vybrant DyeCycle Violet stain after 24 hr in vitro culture. CD11b[pos]Ly6G[pos]CD101[neg] (CD101[neg]) and CD11b[pos]Ly6G[pos]CD101[pos] (CD101[pos]) cells were isolated by cell sorting from ischemic myocardium 24 hr after reperfusion. Data are presented as mean ± SEM (n=3–4). *p<0.05.

(*Figure 7B*). These findings provide evidence that immature CD101[neg] neutrophils are able to traffic into lymph nodes after AMI, where they could contribute to regulation of lymphocyte effector functions.

## Elevated circulating levels of IFN-γ in patients with expanded CD10[neg] neutrophils and increased frequency of CD4[+]CD28[null] T-cells

In order to investigate the potential immunoregulatory effects on T-cells of immature CD10[neg] neutrophils we first performed flow cytometric analysis of CD4[+] T-cells and analyzed circulating levels of IFN-γ in a subgroup of patients. Contrary to some reports, circulating levels of naive (CCR7[+]-CD45RA[+]), central memory (CCR7[+]CD45RA[-]), effector memory (CCR7[-]CD45RA[-]), terminally differentiated effector cells (EMRA, CCR7[-]CD45RA[+]) and CD4[+]CD28[null] T-cells were not significantly different among patients with ACS (*Table 4*).

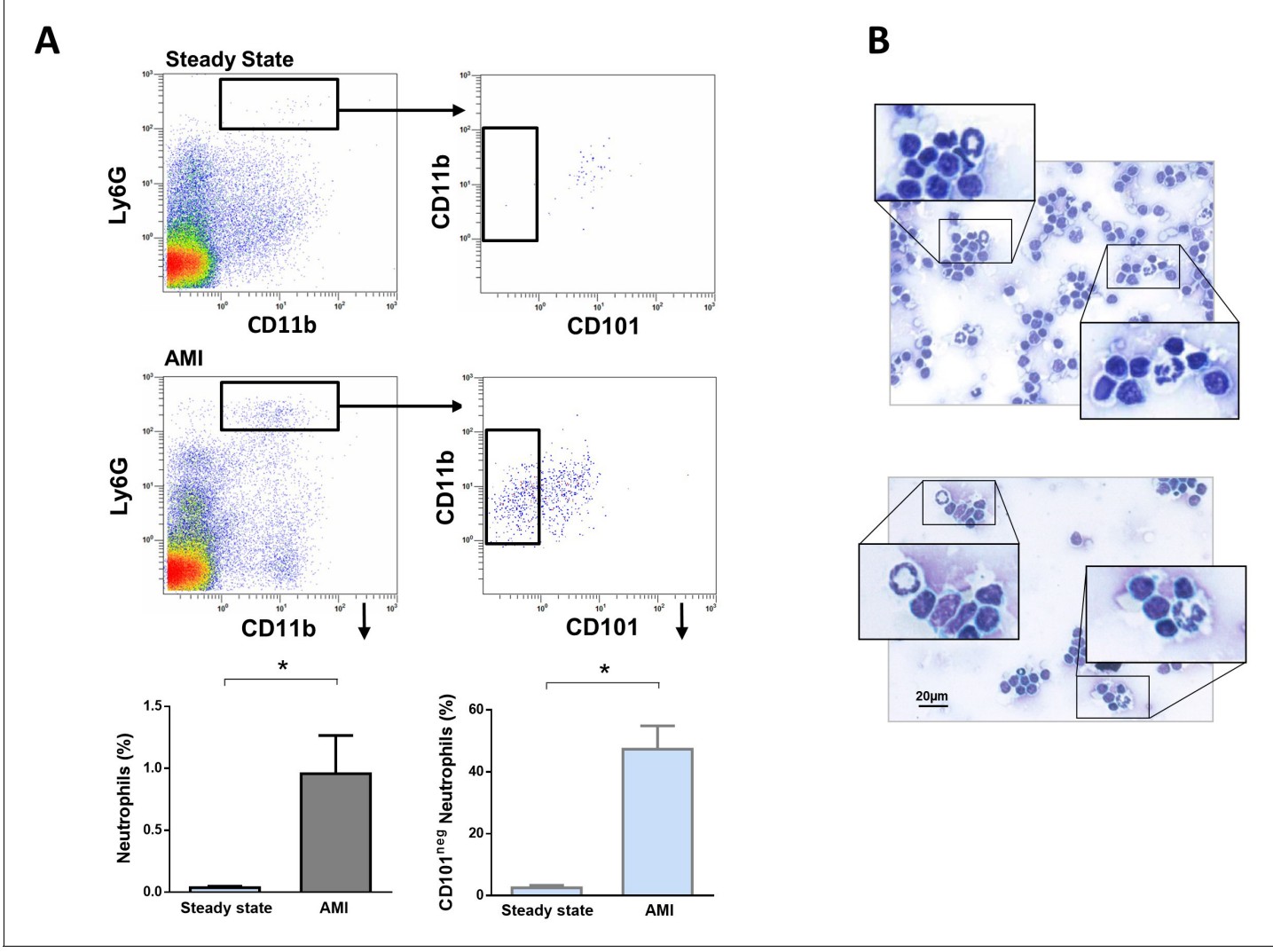

**Figure 7.** Immature CD101[neg] neutrophils are present in mediastinal lymph nodes draining the heart after acute MI. (**A**) Flow cytometric analysis of neutrophil subsets in mediastinal lymph nodes at steady state and 3 hr after reperfusion (AMI). (**B**) Cytospins of mediastinal lymph node cell suspensions stained with Wright's Giemsa. Mice were subjected to 1 hr of coronary occlusion and heart-draining lymph nodes were isolated 3 hr after reperfusion. Data are presented as mean ± SEM (n=3). *p<0.05.

Notably, we found that elevated circulating IFN-γ levels and increased frequency of CD10[neg] neutrophils were linked, mainly in patients with expanded CD4[+]CD28[null] T-cells (*Figure 8A and B*). To better highlight the relationship among IFN-γ, CD10[neg] neutrophils and CD4[+]CD28[null] T-cells we

**Table 4.** CD4[+] T-cells Subsets.

|  | UA (*N*=11) | NSTEMI (*N*=13) | STEMI (*N*=34) | p (K-W) |
|---|---|---|---|---|
| NAIVE (n/μL) | 440 (338-511) | 497 (328-567) | 382 (312-627) | n.s. |
| CM (n/μL) | 300 (233-364) | 278 (242-328) | 334 (210-478) | n.s |
| EM (n/μL) | 122 (91-140) | 114 (86-137) | 102 (80-165) | n.s. |
| EMRA (n/μL) | 55 (30-77) | 42 (24-66) | 47 (29-109) | n.s. |
| CD4[+]CD28[null] (n/μL) | 4 (3–39) | 5 (4–28) | 10 (1-38) | n.s. |

Data are presented as median (IQR). NAIVE, CCR7[+]CD45RA[+]; CM, CCR7[+]CD45RA[-]; EM, CCR7[-]CD45RA[-]; EMRA, CCR7[-]CD45RA[+]. Kruskal-Wallis (K-W) test; n.s., not significant.

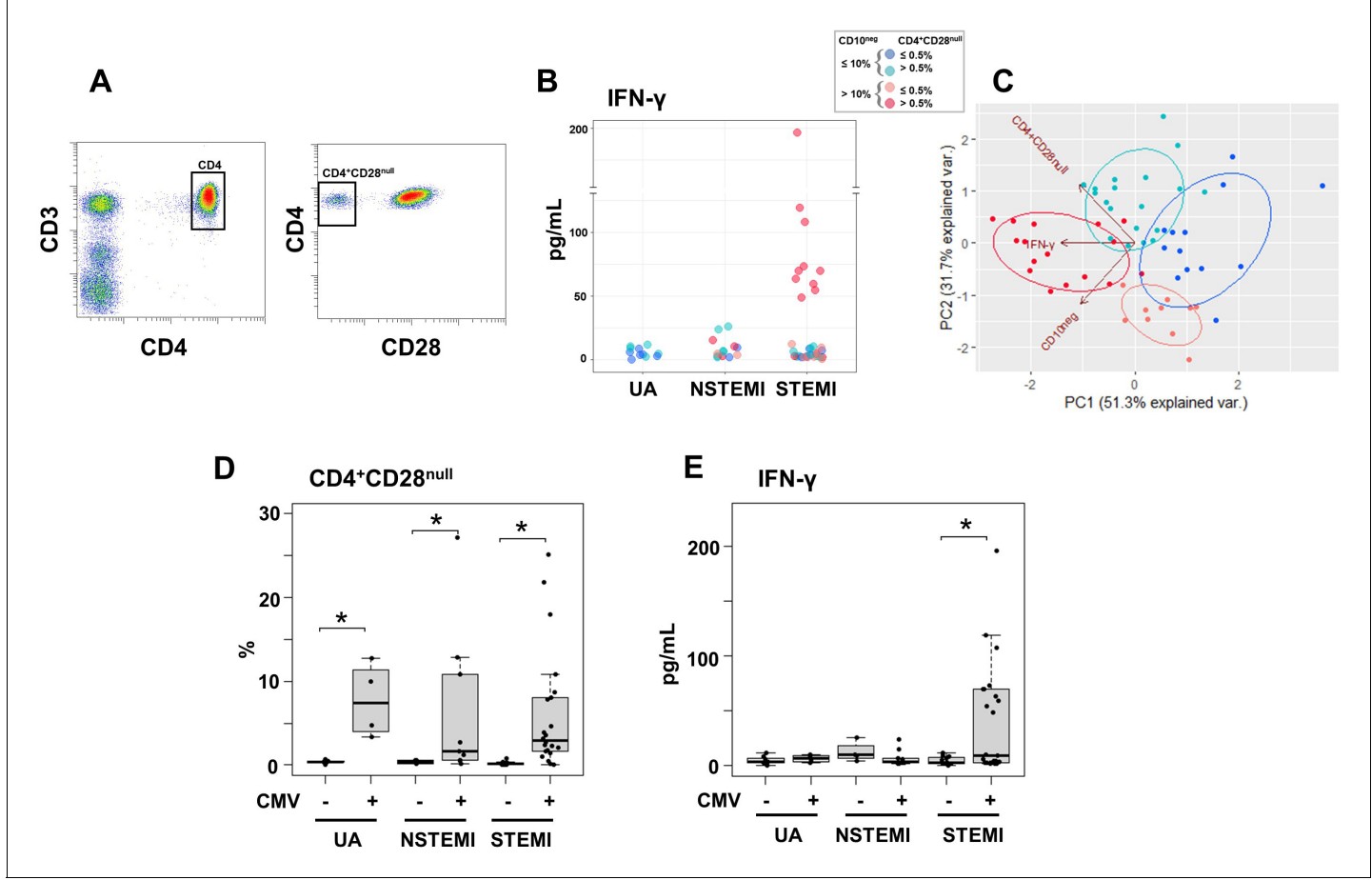

**Figure 8.** Elevated IFN-γ levels in patients with expanded CD10[neg] neutrophils and increased frequency of CD4+CD28[null] T-cells. (**A**) Gating strategy identifying CD4+CD28[null] T-cells. (**B**) Scatter plot showing IFN-γ levels according to frequency of CD10[neg] neutrophils and CD4+CD28[null] T-cells. (**C**) Principal component analysis (PCA) showing clustering according to circulating levels of IFN-γ, CD10[neg] neutrophils and CD4+CD28[null] T-cells. Patients were stratified based on frequency of CD10[neg] neutrophils (≤10% or >10%) and frequency of CD4+CD28[null] T-cells (≤0.5% or >0.5%). (**D**) Frequency of CD4+CD28[null] T-cells in patients with acute coronary syndrome stratified according to cytomegalovirus (CMV) serostatus. (**E**) circulating IFN-γ levels stratified according to CMV serostatus. UA (n=11), NSTEMI (n=13), and STEMI (n=34). *p≤0.05.

The online version of this article includes the following figure supplement(s) for figure 8:

**Figure supplement 1.** (A) CD4+CD28[null] T-cell frequency distribution (log$_{10}$-transformed CD4+ T-cell fractions) of CMV± (top, n=58), CMV- (middle, n=23), and CMV+ (bottom, n=35) acute coronary syndrome (ACS) patients. CD4+CD28[null] T-cells displayed a bimodal distribution related to CMV-seropositivity. (B) Boxplots show the log$_{10}$-transformed frequency of CD4+CD28[null] T-cells in CMV--UA, CMV- (blue) and CMV+ (red) -ACS patients. Expansion index (dotted line) was calculated as UQ+1.5xIQR of CMV--UA patients chosen as reference group. CD4+CD28[null] T-cell frequency more than 0.5% was considered as an index of expansion; UQ (upper quantile), IQR (inter-quantile range). (C) CD10[neg] neutrophils (CD10[neg]) frequency distribution (log$_{10}$-transformed) of CMV± (top, n=71), CMV- (middle, n=31), and CMV+ (bottom, n=40) ACS patients. (D) Boxplots show the log$_{10}$-transformed frequency of CD10[neg] in UA, NSTEMI and STEMI patients. Expansion index (dotted line) was calculated as UQ+1.5xIQR of UA patients. According, patients with CD10[neg] frequency more than 10% had expansion. (E) Scaled frequency of CD4+CD28[null] T-cells and CD10[neg] neutrophils stratified by criteria of cell expansion. Hierarchical clustering performed on columns highlights the relationship among CD10[neg] neutrophils, CD4+CD28[null] T-cells, IFN-γ production and CMV seropositivity.

**Figure supplement 2.** Relationship among the peripheral expansion of immature CD10[neg] neutrophils, CMV IgG titers, CD4+CD28[null] T-cells, and circulating levels of IFN-γ and IL-12 in patients with AMI.

also performed principal component analysis (PCA) that showed clustering according to elevated circulating levels of IFN-γ, high levels of CD10[neg] neutrophils and peripheral expansion of CD4+-CD28[null] T-cells (*Figure 8C*). The highest IFN-γ levels were found in STEMI patients with expanded CD10[neg] neutrophils (>10%) and increased frequency of CD4+CD28[null] T-cells (*Figure 8C*).

Altered T-cell homeostasis and increased frequency of circulating CD28[null] T-cells have been linked to cytomegalovirus (CMV) seropositivity (*Pera et al., 2018*; *Looney et al., 1999*; *Moro-García et al., 2018*). Therefore, we next analyzed the impact of CMV serostatus on CD4+CD28[null]

T-cells frequency. We discovered that CD4$^+$ T-cells lacking the costimulatory molecule CD28 showed expansion across CMV-seropositive (CMV$^+$) patients (*Figure 8D*).

Frequency distribution of CD4$^+$CD28$^{null}$ T-cells (log$_{10}$ transformed to improve visualization) appeared bimodal and analyzing separately in CMV$^+$ and CMV-seronegative (CMV$^-$) patients, the median was significantly higher by a factor 14.1 in CMV$^+$ patients. (*Figure 8—figure supplement 1A*). Moreover, CD4$^+$CD28$^{null}$ frequency positively correlated with CMV-IgG antibody levels (R=0.6, p<10$^{-5}$). Therefore, we believe that the expansion of CD4$^+$CD28$^{null}$ T-cells may not be a direct result of coronary events but appears to be related to the CMV-induced immune changes secondary to repeated antigen exposure.

We defined expansion of CD4$^+$CD28$^{null}$ T-cells frequency a non-parametric, upper outlier limit (upper quartile+1.5×interquartile range) as previously reported (*Pera et al., 2018*). The subgroup of CMV$^-$ patients with unstable angina was considered as reference group. (*Figure 8—figure supplement 1B*). Similarly, we derived a cut-off for expansion of CD10$^{neg}$ neutrophils but taking as reference group the whole cohort of patients with UA since frequency expansion appeared prevalently due to the grade of coronary disease, not to CMV-seropositivity and both contributions to cell expansion could not be dissected (*Figure 8—figure supplement 1C and D*). Then, in order to highlight relationship among IFN-γ levels, CD10$^{neg}$ neutrophils, CD4$^+$ T-cell subsets and CMV-seropositivity we performed hierarchical clustering stratifying patients with ACS according to the expansion cut-offs above described. This individuated four subgroups of patients with frequency of CD10$^{neg}$ neutrophils (≤10% or >10%) and frequency of CD4$^+$CD28$^{null}$ T-cells (≤0.5% or >0.5%) as depicted in *Figure 8C* by principal component analysis (PCA). In the derived heatmap IFN-γ, CD10$^{neg}$ neutrophils, CD4$^+$CD28$^{null}$, EMRA, and EM CD4$^+$ T-cells were grouped together showing similar patterns (*Figure 8—figure supplement 1E*), indicating that persistent CMV infection is associated with expansion of the effector memory CD4$^+$ T-cell compartment and higher IFN-γ levels in patients with increased frequency of CD10$^{neg}$ neutrophils. Not surprisingly, when stratified according to CMV serostatus, maximum levels of circulating IFN-γ among ACS patients were detected in CMV-seropositive STEMI patients displaying increased levels of CD10$^{neg}$ neutrophils (*Figure 8E*), indicating a relation among expansion of immature CD10$^{neg}$ neutrophils, CMV seropositivity and strongly enhanced levels of IFN-γ in patients with large AMI. Of note, PCA performed replacing the percentage of CD4$^+$CD28$^{null}$ cells with CMV-IgG titer (stratified by a cut-off equal to 0.55) showed a better cluster grouping (*Figure 8—figure supplement 2A*), with similar represented total variance (PC1+PC2: 79.5% vs. 83% respectively). The heatmap derived stratifying patients by frequency of CD10$^{neg}$ neutrophils and CMV-IgG cut-offs (*Figure 8—figure supplement 2B*) shows hierarchical subgrouping of CMV-IgG titers and CD4$^+$CD28$^{null}$ cells. Moreover, Spearman's correlation matrix revealed a strong positive correlation between CMV-IgG titers and the percentage of CD4$^+$CD28$^{null}$ cells (*Figure 8—figure supplement 2C*). Taken together, these results highlight the complex relationship in vivo among CMV seropositivity, CD4$^+$CD28$^{null}$ cells, CD10$^{neg}$ neutrophils and IFN-γ levels.

## CD10$^{neg}$ neutrophils via induction of interleukin-12 enhance priming for IFN-γ production by CD4$^+$ T-cells

Environmental factors such as CMV infection can induce changes in CD4$^+$ T-cell phenotype and function. Consequently, to provide a mechanistic understanding of the cellular basis for raised IFN-γ in CMV-seropositive patients with expanded CD10$^{neg}$ neutrophils, we investigated IFN-γ secretion by CD4$^+$ T-cells isolated from CMV$^-$/CMV$^+$ patients and its potential link to interleukin 12 (IL-12), potent inducer of IFN-γ (*Trinchieri, 2003*). In cell-to-cell contact-dependent conditions human neutrophils can mimic myeloid-derived suppressor cells and suppress T-cell activation through artefactual mechanisms (*Negorev et al., 2018*). Therefore, CD10$^{neg}$/CD10$^{pos}$ neutrophils were evaluated for their ability to enhance IFN-γ production in cell contact-independent manner. We found that CD10$^{neg}$ neutrophils strongly enhanced IFN-γ and IL-12 production by CD4$^+$ T-cells from CMV$^+$ patients (*Figure 9A and B*), when co-cultured using a transwell system where CD4$^+$ T-cells in the lower chamber were separated from neutrophils in the upper chamber. Of note, CD4$^+$ T-cells equally responded to cell-free supernatants derived from CD10$^{neg}$ neutrophils. IFN-γ and IL-12 production were significantly higher in CD4$^+$ T-cells from CMV$^+$ than CMV$^-$ patients. The addition of neutralizing anti-IL-12 antibody abrogated the IFN-γ production by CD4$^+$ T-cells from CMV$^+$ patients in presence of supernatants derived from CD10$^{neg}$ neutrophils (*Figure 9A*). Our data indicate that CD10$^{neg}$

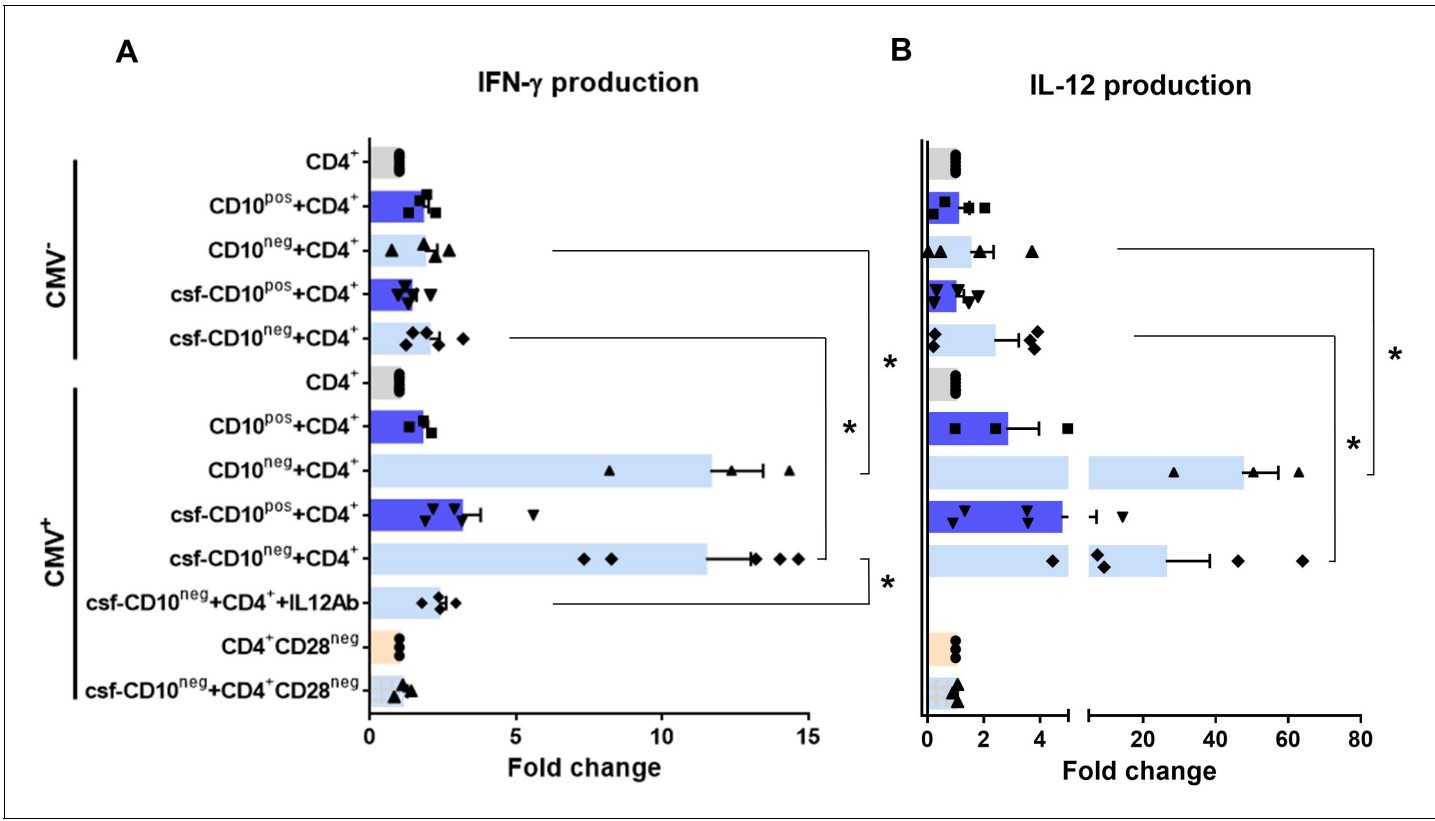

**Figure 9.** CD10[neg] neutrophils enhance IFN-γ production by CD4[+] T-cells via induction of interleukin-12. (**A**) IFN-γ and (**B**) interleukin-12 production by CD4[+] T-cells stimulated with anti-CD3/CD28 beads and co-cultured for 24 hr in absence (CD4[+]) or presence of CD10[pos] neutrophils (CD10[pos]+CD4[+]), CD10[neg] neutrophils (CD10[neg]+CD4[+]) using a transwell system or cultured with cell-free supernatants derived from CD10[pos] neutrophils (csf-CD10[pos]+CD4[+]), CD10[neg] neutrophils (csf-CD10[neg]+CD4[+]), CD10[neg] neutrophils in the presence of neutralizing anti-IL-12 antibody (csf-CD10[neg]+CD4[+]+IL12Ab). CD4[+]CD28[null] T-cells were stimulated with anti-CD3/CD28 beads (CD4[+]CD28[neg]) and cultured with cell-free supernatants derived from CD10[neg] neutrophils (csf-CD10[neg]+CD4[+]CD28[neg]). CD10[neg]/CD10[pos] neutrophils, CD4[+] T-cells and CD4[+]CD28[null] T-cells were isolated from CMV-seronegative (CMV-) or CMV-seropositive (CMV[+]) patients with AMI (n=3–5). Data are represented as fold-change to respective CD3/CD28-stimulated cells and presented as mean ± SEM. *p≤0.05.

The online version of this article includes the following figure supplement(s) for figure 9:

**Figure supplement 1.** IFN-γ production by CD4[+] T-cells from CMV-seropositive AMI patients stimulated with anti-CD3/CD28 beads and cultured for 24 hr in absence (CD4[+]) or presence of cell-free supernatants derived from CD10[pos] neutrophils (csf-CD10[pos]+CD4[+]) or CD10[neg] neutrophils (csf-CD10[neg]+CD4[+]).

neutrophils release soluble factors that efficiently induce a strong Th1 type response. Further studies aiming at characterizing the neutrophil-secreted immunomodulatory factors are ongoing.

Notably, bioinformatic analysis showed enhanced circulating levels of IL-12 in CMV[+] patients with increased frequency of CD10[neg] neutrophils (*Figure 8—figure supplement 2B*). Moreover, Spearman's correlation matrix showed multiple inter-correlations among IL-12 circulating levels, CMV IgG titers, frequency of CD10[neg] neutrophils, expanded CD4[+]CD28[null] T-cells and particularly circulating levels of IFN-γ (*Figure 8—figure supplement 2C*). To test whether neutrophils from CMV- or CMV[+] patients have different ability to induce IFN-γ, CD4[+] T-cells from CMV[+] patients were cultured in presence of supernatants derived from CMV-seronegative patients (*Figure 9—figure supplement 1*). We found that CD10[neg] neutrophil supernatants markedly enhanced IFN-γ secretion (*Figure 9—figure supplement 1*), indicating that CD10[neg] neutrophils from CMV- and CMV[+] patients exhibit the same ability to induce IFN-γ production by CD4[+] T-cells. CD10[neg] neutrophils had no effect on CD3/CD28-stimulated CD4[+]CD28[null] T-cells (*Figure 9A and B*), demonstrating that overproduction of IFN-γ is confined to CD4[+] T-cells expressing CD28. Taken together, our findings indicate that CD4[+]CD28[+] T-cells from CMV[+] patients with AMI display a distinct phenotype overproducing IFN-γ in presence of immature neutrophils via induction of interleukin-12.

## Discussion

Innate immune mechanisms play a paramount role during AMI and the functional heterogeneity of monocytes and neutrophils have been the focus of intensive research in recent years. This study highlights for the first time that immature $CD16^+CD66b^+CD10^{neg}$ neutrophils and $CD14^+$HLA-$DR^{neg/low}$ monocytes promoting proinflammatory immune responses expand in the peripheral blood from patients with large AMI. We also show that immature neutrophils are recruited to the injured myocardium and migrate to mediastinal lymph nodes shortly after reperfusion, using a mouse model of AMI. Furthermore, we found a potential link among increased frequency of immature CD10neg neutrophils and elevated IFN-γ levels, especially in cytomegalovirus-seropositive patients with expanded $CD4^+CD28^{null}$ T-cells. Finally, we could show that $CD10^{neg}$ neutrophils enhance $CD4^+$ T-cells IFN-γ production by a contact-independent mechanism involving IL-12 (*Figure 10*).

This study uncovered that CD10 can be used as a surface marker to identify the immature neutrophil population that expands and promotes proinflammatory effects in patients suffering from AMI. We believe that immature $CD10^{neg}$ neutrophils derive from MI-induced emergency granulopoiesis. Both mature (segmented) and immature banded neutrophils are released from the bone marrow

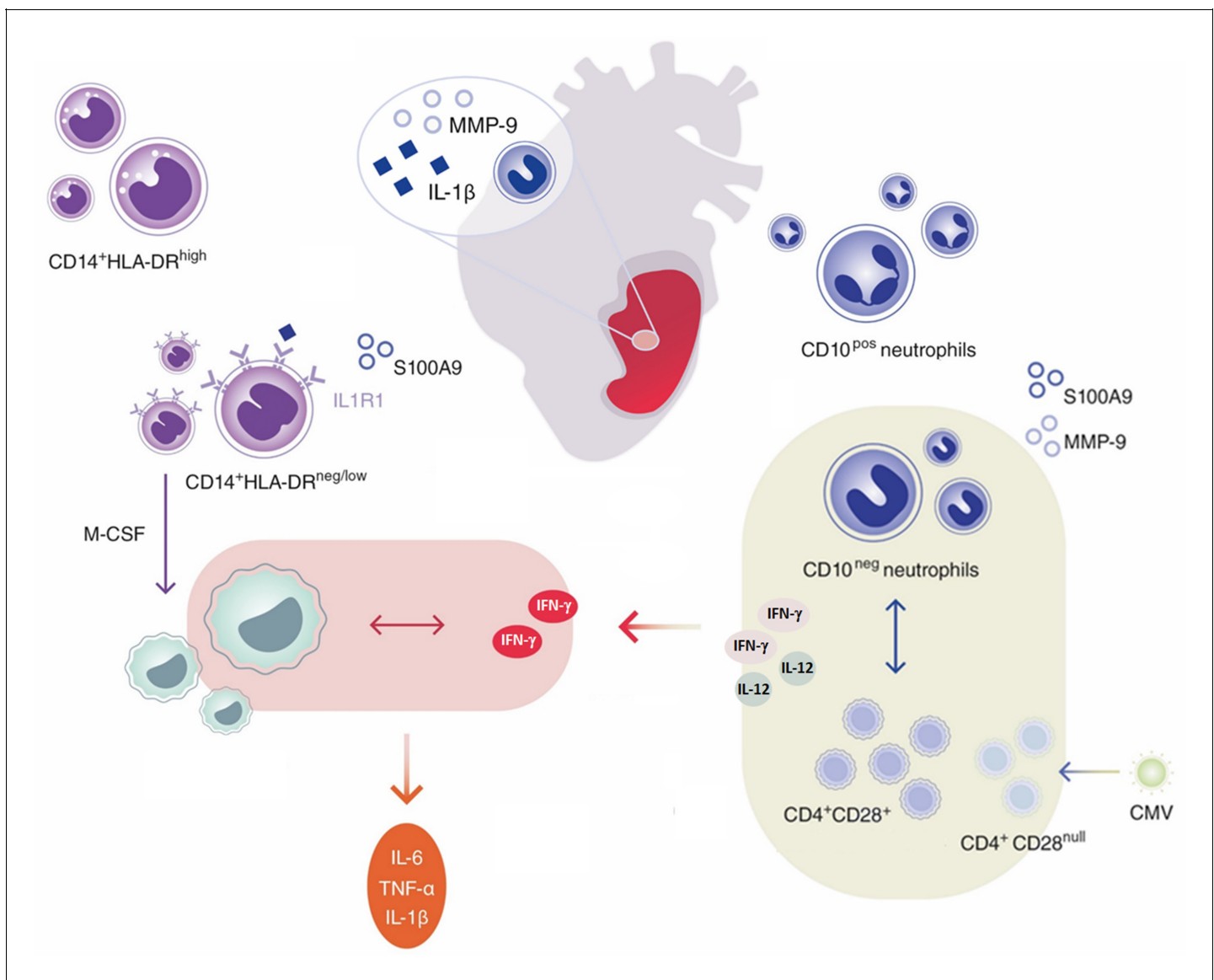

**Figure 10.** Immature $CD10^{neg}$ neutrophils and HLA-$DR^{neg/low}$ monocytes inducing proinflammatory and adaptive immune responses emerge in patients with large acute myocardial infarction.

presumably to meet the high demand for more neutrophils, especially in patients with large AMI. Not surprisingly, in our study higher frequency of circulating CD10$^{neg}$ neutrophils was associated with increased systemic concentrations of G-CSF, an essential regulator of neutrophil trafficking from the bone marrow to the blood and important neutrophil survival factor (*Costa et al., 2019*; *Silvestre-Roig et al., 2020*). Recently, CD10 has been proposed as a marker that distinguishes mature from immature neutrophils in healthy volunteers receiving G-CSF for stem cell mobilization (*Marini et al., 2017*).

Multiple clinical trials have evaluated the use of G-CSF in patients with AMI after successful revascularization. The majority of these studies found that effective stem cell mobilization with G-CSF therapy failed to improve left ventricular recovery (*Traverse, 2019*). Our findings suggest that the therapeutic benefits of G-CSF therapy after AMI might be compromised due to the release of immature proinflammatory CD10$^{neg}$ neutrophils.

However, neutrophils may be released from the bone marrow in response to increased damage-associated molecular patterns such as S100A8/S100A9, secreted from neutrophils as mediators of sterile inflammation (*Sreejit et al., 2020*). Of interest, we found that circulating CD10$^{neg}$ neutrophils express high amounts of S100A9, indicating that immature neutrophils could be an important source of this alarmin in patients with AMI.

Under inflammatory conditions neutrophils traffic to inflamed tissues as well as to draining lymph nodes (*Costa et al., 2019*; *Leliefeld et al., 2015*) modulating T cell-mediated immune responses. Our data from the mouse model of AMI are first to provide evidence that immature neutrophils can populate heart-draining lymph nodes in response to acute ischemic injury. It is intriguing to speculate that CD10$^{neg}$ neutrophils migrate into lymph nodes where they could encounter T-cells and shape adaptive immune responses.

Emerging evidence indicates that immature neutrophils can be T-cell suppressive or do possess T-cell stimulatory capacities, displaying disease-specific functional plasticity (*Costa et al., 2019*; *Rahman et al., 2019*). Immunostimulatory immature CD10$^{neg}$ neutrophils appear in the circulation of G-CSF–treated healthy volunteers and contact-dependent mechanisms account for their immunoregulatory functions (*Marini et al., 2017*). Here, we provide mechanistic evidence that immature CD10$^{neg}$ neutrophils from patients with AMI, in a contact-independent way involving IL-12, enhance priming for IFN-γ production in activated CD4$^+$ T-cells. Thus, through diverse mechanisms immature CD10$^{neg}$ neutrophils may exert immunostimulatory/proinflammatory functions actively participating in the regulation of adaptive immunity.

Genetic and environmental factors shape the immune system over time. Several studies have demonstrated that persistent CMV infection is associated with changes in T-cell phenotype and function (*Nikolich-Zugich, 2008*; *Wertheimer et al., 2014*; *Davenport et al., 2020*). Our results highlight that CD4$^+$CD28$^+$ T-cells from CMV-seropositive AMI patients are skewed toward a Th1 phenotype, producing large amounts of IFN-γ in presence of CD10$^{neg}$ neutrophils. However, results obtained in vitro cannot be translated directly to the in vivo situation and several cellular and molecular mechanisms could have led to increased circulating levels of the pleiotropic cytokine IFN-γ after AMI. Notably, using bioinformatic tools (PCA and hierarchical clustering) we were able to highlight the tight relationship among the peripheral expansion of immature CD10$^{neg}$ neutrophils, CMV-altered CD4$^+$ T-cell homeostasis and high levels of IFN-γ in patients with large AMI. Thus, determination of circulating CD10$^{neg}$ neutrophils levels, particularly in the context of persistent CMV infection, might help to identify patients at risk for excessive inflammatory immune response.

Although a pathogenetic role of CD4$^+$CD28$^{null}$ T-cells in coronary artery disease and atherogenesis have been recognized, important issues have remained unresolved (*Dumitriu et al., 2009*). A recent study revealed complex associations between of CD4$^+$CD28$^{null}$ T-cells and cardiovascular disease (*Tomas et al., 2020*). CD4$^+$CD28$^{null}$ T-cells are associated with a lower risk for first-time coronary events in a population-based cohort. In contrast, in patients with advanced atherosclerotic disease an increased frequency of CD4$^+$CD28$^{null}$ T-cells was associated with more frequent major adverse cardiovascular events (*Tomas et al., 2020*). Our findings point to a potential link between CMV induced immune alterations following repeated antigen exposure and the peripheral expansion of CD4$^+$CD28$^{null}$ T-cells in ACS patients. CMV has been associated with atherosclerosis and increased risk for cardiovascular diseases. Recent clinical data showed that myocardial ischemia in CMV-seropositive patients leads to significant changes in the composition of the CD8$^+$ T-cell repertoire, accelerating immunosenescence (*Hoffmann et al., 2015*).

In spite of numerous studies on polymorphonuclear myeloid cells the presence and functional characteristics of immature neutrophils is underexplored in the setting of AMI in mice. The present study demonstrated for the first time that CD101 can be used as a marker to define the maturation status of neutrophils mobilized into the peripheral blood in response to ischemia and recruited to sites of ischemic injury after reperfusion. Previous studies in a human model of experimental endotoxemia showed that banded neutrophils exhibit efficient migration to sites of infection (*van Grinsven et al., 2019*). Moreover, developmental analysis of bone marrow neutrophils revealed that immature neutrophils possess functional migratory machinery and are recruited to the periphery of tumor-bearing mice (*Evrard et al., 2018*). Of note, we found that immature CD101$^{neg}$ neutrophils are released into the bloodstream within minutes after reperfusion and are capable of efficient migration to ischemic tissues, displaying increased expression of MMP-9 and IL-1ß at 3 and 24 hr after reperfusion, respectively. There are significant differences between mouse and human immunology. In mice, 10–25% of circulating leukocytes are neutrophils, while in humans neutrophils are the predominant leukocyte population, accounting for 50–70% of leukocytes. Moreover, the transit time of leukocytes may be quite different (*Mestas and Hughes, 2004*; *Mackey et al., 2019*; *Riehle and Bauersachs, 2019*). During homeostasis, trafficking of neutrophils/myeloid cells from bone marrow into the circulation takes between 1–2 days in mice and 5–8 days in humans (*Mackey et al., 2019*). Such differences should be considered when comparing animal and human studies on immune mechanisms underlying wound healing. However, future studies that employ intravital microscopy are required to track the fate of immature neutrophils mobilized from the bone marrow and recruited to the ischemic myocardium as well as lymph nodes after AMI.

The recruitment of immune cells to sites of tissue repair is a complex highly regulated process involving cytokines, chemokines, and interactions between infiltrating immune cells. HLA-DR$^{neg/low}$ monocytes from patients with AMI are not immunosuppressive but express high amounts of IL1R1. Thus, immature neutrophils, as an important source of IL-1ß in the reperfused heart, may be actively involved in the recruitment of HLA-DR$^{neg/low}$ cells. *Saxena et al., 2013* showed that IL1R1 signaling mediates early recruitment of Ly6C$^{hi}$ monocytes to the infarcted myocardium. Reperfused myocardial infarction had intense infiltration with Ly6C$^{hi}$ monocytes expressing IL1R1 that peaked after 24 hr of reperfusion (*Saxena et al., 2013*). Noteworthy, recent studies demonstrated that the failing human heart also contains HLA-DR$^{neg/low}$ monocytes (*Bajpai et al., 2018*).

Several immune mechanisms operate during cardiac wound healing and IFN-γ plays different roles depending on the cellular and microenvironmental context intrinsically linked to the stages of ischemic injury. By integrating cell sorting and in vitro experiments, we found that macrophages differentiated from HLA-DR$^{neg/low}$ monocytes produced more TNF-α, IL-6, and IL-1ß upon IFN-γ stimulation as HLA-DR$^{high}$ monocyte-derived macrophages. These findings may support a role for HLA-DR$^{neg/low}$ monocytes in pathogenic mechanisms operating during AMI and may, at least in part, explain why an expansion of circulating HLA-DR$^{neg/low}$ monocytes correlates with circulating levels of TNF-α, IL-6, and IL-1ß.

The interleukin-1 pathway plays a key role in post-MI inflammation and the progression to heart failure (*Abbate et al., 2020*). Our in vitro mechanistic experiments with immune cells from AMI patients as well as mouse studies provide a potential linkage between the induction of immature CD10$^{neg}$ neutrophils/HLA-DR$^{neg/low}$ monocytes and increased interleukin-1 activity during AMI. Emerging evidences highlight that targeting interleukin-1 may hold promise for patients after MI (*Buckley and Abbate, 2018*). In STEMI patients, the interleukin-1 receptor antagonist anakinra significantly reduced the systemic inflammatory response. Moreover, in the CANTOS trial, administration of canakinumab (a monoclonal antibody targeting IL-1β) prevented the recurrence of ischemic events, reduced heart failure-related hospitalizations and mortality in patients with prior AMI (*Buckley and Abbate, 2018*).

Neutrophils may orchestrate post-MI healing through induction of MerTK expression and increased efferocytosis capacity of macrophages (*Horckmans et al., 2017*). Of interest, we found significantly higher expression of MerTK on macrophages differentiated from CD14$^{+}$HLA-DR$^{neg/low}$ monocytes and polarized in the presence of secretome derived from CD10$^{pos}$ neutrophils. It is tempting to speculate that HLA-DR$^{neg/low}$ monocytes had the potential to differentiate into macrophages with enhanced capacity to phagocyte apoptotic cells when exposed to CD10$^{pos}$ neutrophils after AMI. Nevertheless, in vitro studies have limited ability to explain the complexity of macrophage activation within the infarct environment.

The apoptosis rate of CD10$^{neg}$ neutrophils was strongly reduced compared with CD10$^{pos}$ neutrophils. This finding is consistent with recent studies showing that cellular immaturity influences apoptosis (*Marini et al., 2017*; *Wang et al., 2020*). We also observed that CD101$^{neg}$ neutrophils sorted from reperfused ischemic hearts were resistant to apoptosis. Apoptosis is essential for neutrophil functional shutdown, phagocytic clearance by macrophages and timely resolution of inflammation (*Koedel et al., 2009*; *Elliott et al., 2017*). Therefore, apoptosis resistance of immature neutrophils may play a crucial role, hitherto overlooked, after MI. Further depletion/modulation studies in mice will shed light on the immunoregulatory functions of immature neutrophils in the inflammatory response during wound healing.

In conclusion, this study shows that immature CD10$^{neg}$ neutrophils and CD14$^{+}$HLA-DR$^{neg/low}$ monocytes expand in patients with AMI and highlights their potential role as triggers of immune/inflammatory dysregulation after ischemic injury. These findings could have major implications for understanding immunoregulatory mechanisms operating during AMI and for the development of future therapeutic strategies. Nevertheless, further studies deciphering the relationship between elevated CD14$^{+}$HLA-DR$^{neg/low}$ monocytes/CD10$^{neg}$ neutrophils and ensuing mortality and morbidity after ischemic injury are necessary and ongoing.

## Acknowledgements

We are grateful for the support of Dr Matthias Ballmaier from the Central Research Facility Cell Sorting of the Hannover Medical School. D Fraccarollo and J Bauersachs received support from the Deutsche Forschungsgemeinschaft (BA 1742/8-1).

## Additional information

### Funding

| Funder | Grant reference number | Author |
| --- | --- | --- |
| Deutsche Forschungsgemeinschaft | BA 1742/8-1 | Daniela Fraccarollo Johann Bauersachs |

The funders had no role in study design, data collection and interpretation, or the decision to submit the work for publication.

### Author contributions

Daniela Fraccarollo, Conceptualization, Supervision, Funding acquisition, Validation, Investigation, Visualization, Methodology, Writing - original draft, Project administration; Jonas Neuser, Resources, Investigation, Writing - review and editing; Julian Möller, Investigation; Christian Riehle, Resources, Writing - review and editing; Paolo Galuppo, Data curation, Software, Formal analysis, Validation, Investigation, Visualization, Writing - original draft; Johann Bauersachs, Resources, Supervision, Funding acquisition, Project administration, Writing - review and editing

### Author ORCIDs

Daniela Fraccarollo https://orcid.org/0000-0002-7756-6032
Julian Möller http://orcid.org/0000-0003-1818-6782

### Ethics

Human subjects: The study protocol is in accordance with the ethical guidelines of the 1975 declaration of Helsinki and has been approved by the local ethics committee of Hannover Medical School. Patients referred to our department for acute coronary syndrome (ACS) were included after providing written informed consent.
Animal experimentation: All animal experiments were conducted in accordance with the Guide for the Care and Use of Laboratory Animals published by the National Institutes of Health (Publication No. 85-23, revised 1985). All procedures were approved by the Regierung von Unterfranken (Würzburg, Germany; permit No. 54-2531.01-15/07) and by the Niedersächsisches Landesamt für

Verbraucherschutz und Lebensmittelsicherheit (Oldenburg, Germany; permit No. 33.12-42502-04-11/0644; 33.9-42502-04-13/1124 and 33.12-42502-04-17/2702).

## Decision letter and Author response
Decision letter https://doi.org/10.7554/eLife.66808.sa1
Author response https://doi.org/10.7554/eLife.66808.sa2

## Additional files

### Supplementary files
• Transparent reporting form

### Data availability

All data generated or analysed during this study are included in the manuscript and supporting files. Source data files have been provided for Figure 1-figure supplement 1 and Figure 3-figure supplement 1.

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

# Appendix 1

**Appendix 1—key resources table**

| Reagent type (species) or resource | Designation | Source or reference | Identifiers | Additional information |
|---|---|---|---|---|
| biological sample (*Homo sapiens*) | Peripheral Blood Leukocytes | Medizinische Hochschule Hannover | | Freshly isolated from Human |
| biological sample (*Mus musculus*) | Peripheral Blood, Heart, Lymph Node, Bone Marrow Leukocytes | Medizinische Hochschule Hannover | | Freshly isolated from *Mus musculus* |
| antibody | anti-human CD14-APC-H7 (Mouse monoclonal) | BD Biosciences | Cat# 560180 RRID:AB_1645464 | 1:50 |
| antibody | anti-human HLA-DR-FITC (Mouse monoclonal) | BioLegend | Cat# 307604 RRID:AB_314682 | 1:30 |
| antibody | anti-human HLA-DR-PE (Mouse monoclonal) | BioLegend | Cat# 307606 RRID:AB_314684 | 1:30 |
| antibody | anti-human HLA-DR-PerCP/Cy5.5 (Mouse monoclonal) | BioLegend | Cat# 307630 RRID:AB_893567 | 1:30 |
| antibody | anti-human CD16- BV605 (Mouse monoclonal) | BioLegend | Cat# 302040 RRID:AB_2562990 | 1:50 |
| antibody | anti-human CX3CR1-PE (Mouse monoclonal) | BioLegend | Cat# 341604 RRID:AB_1595456 | 1:50 |
| antibody | anti-human CCR2-PerCP/Cy5.5 (Mouse monoclonal) | BioLegend | Cat# 357204 RRID:AB_2562004 | 1:50 |
| antibody | anti-human CD66b-FITC (Mouse monoclonal) | BioLegend | Cat# 305104 RRID:AB_314496 | 1:30 |
| antibody | anti-human CD10-BV421 (Mouse monoclonal) | BioLegend | Cat# 312218 RRID:AB_2561833 | 1:20 |
| antibody | anti-human CD10-PE (Mouse monoclonal) | BioLegend | Cat# 312203 RRID:AB_314914 | 1:20 |
| antibody | anti-human CD3-APC-H7 (Mouse monoclonal) | BD Biosciences | Cat# 56076 RRID:AB_1645475 | 1:30 |
| antibody | anti-human CD4-APC (Mouse monoclonal) | BD Biosciences | Cat# 561840 RRID:AB_10895807 | 1:30 |
| antibody | anti-human CD28-BV421 (Mouse monoclonal) | BioLegend | Cat# 302930 RRID:AB_2561910 | 1:30 |
| antibody | anti-human CCR7-Alexa Fluor 488 (Mouse monoclonal) | BioLegend | Cat# 353206 RRID:AB_10916389 | 1:30 |
| antibody | anti-human CD45RA-PE (Mouse monoclonal) | BioLegend | Cat# 304108 RRID:AB_314412 | 1:30 |
| antibody | anti-human CD114-PE (Mouse monoclonal) | BioLegend | Cat# 346106 RRID:AB_2083867 | 1:25 |

*Continued on next page*

*Appendix 1—key resources table continued*

| Reagent type (species) or resource | Designation | Source or reference | Identifiers | Additional information |
|---|---|---|---|---|
| antibody | anti-human CD177-FITC (Mouse monoclonal) | BioLegend | Cat# 315804 RRID:AB_2072603 | 1:25 |
| antibody | anti-human CD11b-PerCP/Cy5.5 (Mouse monoclonal) | BioLegend | Cat# 301328 RRID:AB_10933428 | 1:25 |
| antibody | anti-human CD101-PE (Mouse monoclonal) | BioLegend | Cat# 331011 RRID:AB_2716106 | 1:25 |
| antibody | anti-human MERTK-APC (Mouse monoclonal) | BioLegend | Cat# 367612 RRID:AB_2687289 | 1:25 |
| antibody | anti-human CD163-FITC (Mouse monoclonal) | BD Biosciences | Cat# 563697 RRID:AB_2738379 | 1:25 |
| antibody | anti-IL-12 | R and D Systems | Cat# MAB219 RRID:AB_2123616 | 4 µg/mL |
| antibody | Isotype control | R and D Systems | Cat# MAB002 RRID:AB_357344 | 4 µg/mL |
| antibody | CD16/CD32 Mouse BD Fc Block RUO (Rat monoclonal) | BD Biosciences | Cat# 553142 RRID:AB_394657 | 1:50 |
| antibody | anti-mouse CD45-PerCP/Cy5.5 (Mouse monoclonal) | BioLegend | Cat# 109828 RRID:AB_893350 | 1:100 |
| antibody | anti-mouse CD45-PerCP/Cy5.5 (Rat monoclonal) | BD Biosciences | Cat# 550994 RRID:AB_394003 | 1:100 |
| antibody | anti-mouse F4/80-BV421 (Rat monoclonal) | BD Biosciences | Cat# 565411 RRID:AB_2734779 | 1:100 |
| antibody | anti-mouse F4/80-APC (Rat monoclonal) | BioLegend | Cat# 123116 RRID:AB_893481 | 1:100 |
| antibody | anti-mouse CD11b-Super Bright 600 (Rat monoclonal) | eBioscience | Cat# 63-0112-82 RRID:AB_2637408 | 1:100 |
| antibody | anti-mouse CD11b- PerCP/Cy5.5 (Rat monoclonal) | BD Biosciences | Cat# 550993 RRID:AB_394002 | 1:100 |
| antibody | anti-mouse CD4-Alexa Fluor 488 (Rat monoclonal) | BioLegend | Cat# 100423 RRID:AB_389302 | 1:100 |
| antibody | anti-mouse CD115-Brilliant Violet 605 (Rat monoclonal) | BioLegend | Cat# 135517 RRID:AB_2562760 | 1:100 |
| antibody | anti-mouse CD115-Alexa Fluor 488 (Rat monoclonal) | BioLegend | Cat# 135512 RRID:AB_11218983 | 1:100 |
| antibody | anti-mouse Ly-6G-APC (Rat monoclonal) | BioLegend | Cat# 127613 RRID:AB_1877163 | 1:100 |

*Appendix 1—key resources table continued*

| Reagent type (species) or resource | Designation | Source or reference | Identifiers | Additional information |
|---|---|---|---|---|
| antibody | anti-mouse Ly-6G-PE (Rat monoclonal) | BD Biosciences | Cat# 551461 RRID:AB_394208 | 1:200 |
| antibody | anti-mouse Ly-6G-APC-H7 (Rat monoclonal) | BD Biosciences | Cat# 565369 RRID:AB_2739207 | 1:100 |
| antibody | anti-mouse CXCR2-Alexa Fluor 647 (Mouse monoclonal) | BioLegend | Cat# 149604 RRID:AB_2565563 | 1:100 |
| antibody | anti-mouse CXCR2-PE (Rat monoclonal) | BioLegend | Cat# 149609 RRID:AB_2565689 | 1:100 |
| antibody | anti-mouse CD101-Alexa Fluor 647 (Rat monoclonal) | BD Biosciences | Cat# 564473 RRID:AB_2738821 | 1:100 |
| antibody | anti-mouse CD101-PE (Rat monoclonal) | eBioscience | Cat# 12-1011-80 RRID:AB_1210729 | 1:100 |
| antibody | anti-MMP-9 (Goat polyclonal) | R and D Systems | Cat# AF909 RRID:AB_355706 | 1:100 |
| antibody | anti IL-1ß (Rabbit polyclonal) | Abcam | Cat# ab9722 RRID:AB_308765 | 1:100 |
| antibody | anti-goat IgG (H+L) Cross-Adsorbed Secondary Antibody Alexa Fluor 488 (Donkey polyclonal) | Invitrogen | Cat# A-11055 RRID:AB_2534102 | 1:500 |
| antibody | anti-rabbit IgG (H+L) Highly Cross-Adsorbed Secondary Antibody Alexa Fluor 488 (Goat polyclonal) | Invitrogen | Cat# A-11034 RRID:AB_2576217 | 1:500 |
| commercial assay or kit | MACSxpress Whole Blood Neutrophil Isolation | Miltenyi Biotec | Cat# 130-104-434 | |
| commercial assay or kit | PrepEase RNA Spin | Affymetrix | Cat# PN78766 | |
| commercial assay or kit | RNeasy Plus Mini | QIAGEN | Cat# 74134 | |
| commercial assay or kit | Dynabeads Untouched Human T-cells | Invitrogen | Cat# 11344D | |
| commercial assay or kit | CellTrace Violet Cell Proliferation | Invitrogen | Cat# C34571 | |
| commercial assay or kit | OpTmizer CTS T-Cell Expansion | Gibco | Cat# A1048501 | |
| commercial assay or kit | Dynabeads Human T-Activator CD3/CD28 | Gibco | Cat# 11131D | |
| commercial assay or kit | VybrantDyeCycle Violet/SYTOX AADvanced Apoptosis | Invitrogen | Cat# A35135 | |
| commercial assay or kit | Apotracker Green | Biolegend | Cat# 427402 | |
| commercial assay or kit | MojoSort Human CD4 T Cell Isolation | BioLegend | Cat# 480009 | |
| commercial assay or kit | CD28 MicroBead | Miltenyi Biotec | Cat# 130-093-247 | |

*Continued on next page*

*Appendix 1—key resources table continued*

| Reagent type (species) or resource | Designation | Source or reference | Identifiers | Additional information |
|---|---|---|---|---|
| commercial assay or kit | LEGENDplex | BioLegend | Cat# 740180 | |
| commercial assay or kit | LEGENDplex | BioLegend | Cat# 740589 | |
| commercial assay or kit | LEGENDplex | BioLegend | Cat# 740929 | |
| commercial assay or kit | CMV-IgG-ELISA PKS | Medac Diagnostika | Cat# 115-Q-PKS | |
| commercial assay or kit | Human IFN-gamma Quantikine ELISA | R and D Systems | Cat# DIF50 | |
| commercial assay or kit | iScript Reverse Transcription Supermix | Bio-Rad | Cat# 1708840 | |
| commercial assay or kit | SsoAdvanced Universal SYBR Green Supermix | Bio-Rad | Cat# 172–5271 | |
| chemical compound, drug | VersaLyse Lysing Solution | Beckman Coulter | Cat# A09777 | |
| chemical compound, drug | Ficoll-Paque Premium | SIGMA | Cat# GE17-5442-02 | |
| chemical compound, drug | ß-mercaptoethanol | Sigma Aldrich | Cat# 63689 | |
| chemical compound, drug | Heat Inctivated Fetal Bovin Serum | Gibco | Cat# A3840001 | |
| chemical compound, drug | May Grünwald Stain Solution | Polysciences | Cat# 24981–1 | |
| chemical compound, drug | May Grünwald Giemsa Phosphate Buffer | Polysciences | Cat# 25032–1 | |
| chemical compound, drug | Wright Giemsa Stain Solution | Polysciences | Cat# 24985–1 | |
| chemical compound, drug | Wright Giemsa Stain Phosphate Buffer | Polysciences | Cat# 24984–1 | |
| chemical compound, drug | Wright-Giemsa Stain Set | Astral Diagnostics | Cat# 5585 | |
| chemical compound, drug | Recombinant Human M-CSF Protein | R and D Systems | Cat# 216-MC-005 | 20 ng/mL |
| chemical compound, drug | Recombinant Human IFN-gamma Protein | R and D Systems | Cat# 285-IF | 20 ng/mL |
| chemical compound, drug | Recombinant Human IL-4 | Miltenyi Biotec | Cat# 130-093-920 | 20 ng/mL |
| chemical compound, drug | Dexamethasone | Sigma Aldrich | Cat# D1756 | 1 µM |
| chemical compound, drug | PenStrep | Gibco | Cat# 15140122 | |
| chemical compound, drug | L-Glutamine | Gibco | Cat#25030081 | |
| chemical compound, drug | Human S100A9 | Invitrogen | Cat# A42590 | 2 ng/mL |
| chemical compound, drug | RBC Lysis Buffer | BioLegend | Cat# 420301 | |
| chemical compound, drug | Liberase | Roche Diagnostics | Cat# 5466202001 | |

*Appendix 1—key resources table continued*

| Reagent type (species) or resource | Designation | Source or reference | Identifiers | Additional information |
|---|---|---|---|---|
| chemical compound, drug | Cytofix/Cytoperm Fixation/ Permeabilization Kit | BD Biosciences | Cat# 554714 | |
| other | ChemoTxDisposable Chemotaxis System | Neuro Probe | Cat# NRP-106–8 | |
| other | Transwell Inserts | Thermo Scientific | Cat# 140620 | |

