## [Decision Letter]

**Acceptance summary:**

In this clinical study the authors demonstrated for the first time mobilization of immature neutrophils (CD10^neg^ in humans, CD101^neg^ in mice) following myocardial infarction. The findings in patients are supported by in vitro co-culture experiments and animal studies indicating that these cells are pro-inflammatory and enhance IFN-γ production by T cells. These findings add to the understanding how white blood cells become activated in patients with coronary artery disease following myocardial infarction and contribute to inflammatory activation.

**Decision letter after peer review:**

Thank you for submitting your manuscript "Expansion of CD10^neg^ neutrophils and HLA-DR^neg/low^ monocytes driving inflammatory responses after myocardial infarction" for consideration by *eLife*. Your article has been reviewed by 3 peer reviewers, one of whom is a member of our Board of Reviewing Editors, and the evaluation has been overseen by a Senior Editor. The reviewers have opted to remain anonymous.

The reviewers and Editors have discussed the reviews with one another, and this letter is to help you prepare a revised submission.

Essential Revisions:

1) Figures 1, supplement 1 and Figure 3 supplement 1 show the variation in the percentage of monocyte subpopulations and neutrophil CD10neg over the course of 5 days after the onset of the symptoms. Showing absolute numbers would provide information about the possible mobilization of these cells from the bone marrow or other reservoirs as suggested by the authors.

2) Do CD14+HLA-DRneg/low monocytes together with CD16+CD66b+CD10neg neutrophils improve ROC values compared to single parameters alone?

3) CMV+ patients exhibit an increased prevalence of CD4^+^CD28null T cells. The authors show that in these patients CD4^+^CD28null T cells correlate with levels of IFNγ and high levels of neutrophils CD10neg in CMV+ positive patients, hence providing a link between these three parameters. However, mechanistically, immature neutrophils only stimulate CD4^+^CD28+ T cells to produce IFNγ. What is the relation then between CD4^+^CD28null T cells and immature neutrophils? What is the link the authors want to do here? These discrepancies should be discussed.

4) Where do the authors think the interaction between neutrophils CD10neg and lymphocytes occurs? In the circulation? In the infarcted heart? Do the authors see increased IL-12 levels in plasma from patients with increased amounts of neutrophils CD10neg? Also, the authors should discuss this potential interaction.

5) Is there only a relationship between CMV+ patients and CD4^+^CD28null? In Figure 7, supplement 1 panel C seems not to be a clear correlation between CMV positivity and neutrophil CD10neg numbers. Is this the case? For instance, with CMV-IgG? If not in numbers, how CMV positivity affects neutrophil CD10neg phenotype? Do neutrophil CD10neg isolated from patients CMV+ or CMV- have different ability to induce IFNγ in T-cells?

6) The authors focus on the interaction of immature neutrophils and the adaptive immune system, but multiple studies associate neutrophil activity with monocyte/macrophage functionality. Is there any relation between CD10neg neutrophils and the observed phenotype of monocytes? Is the alarmin S100A9 relevant for monocyte chemotaxis?

7) There are major discrepancies between mouse and human immature monocytes and neutrophils. The authors should discuss it. The samples analyzed in mice for RNAseq are total FACS-sorted neutrophils. Do the authors see more similarities when analyzing CD101- mouse neutrophils?

8) How are the levels of immature neutrophils altered in the bone marrow of mice after ischemia-reperfusion? The authors should also provide absolute numbers of CD101+ and CD101- neutrophils within the ischemic heart to evaluate their contribution.

9) In some cases, conclusions are not supported by robust data. For example, the authors conclude that CD14+HLA-DRneg/lo monocytes play a crucial role in post-infarction inflammation based exclusively on in vitro experiments. Moreover, conclusions regarding the pro-inflammatory role of immature neutrophils are based on in vitro data and associative studies. Please tone down the conclusions and discuss the limitations related to the absence of in vivo experiments depleting immature neutrophils.

10) What is the fate of immature neutrophils in the infarct? These cells are typically short-lived. Did they rapidly undergo apoptosis? Did they mature rapidly within the infarct environment? The mouse model may allow the authors to address these questions.

11) The authors need to explain the rationale for assessment of specific inflammatory genes. For example, in neutrophils, what was the basis for selection of these specific genes (MMP9, IL1R1, IL1R2, STAT3 etc), vs. other inflammatory genes known to be expressed at high levels by neutrophils?

12) The rationale for the use of CD101 as a marker differentiating between mature and immature neutrophils in the mouse needs to be better presented. It seems that the authors identified CD101 as a neutrophil gene that is downregulated following ischemia. On that basis, they used CD101 as a maturation marker. However, why does this suggest that CD101 selectively labels mature vs immature neutrophils? Perhaps its downregulation reflects neutrophil activation, rather than immaturity. Perhaps the use of CD101 as a marker for mature neutrophils in reference 14 may provide a better rationale.

13) The rationale for the study of CMV seropositivity is not well-explained. The data show infiltration of the infarcted heart with immature neutrophils and CD14+HLA-DRneg monocytes. One would have anticipated experiments investigating the (proposed) role of these cells in the post-infarction inflammatory response. The focus on CMV+ patients needs to be better introduced in the manuscript.

14) What was the rationale for the experiment examining interactions of CD10neg neutrophils with T cells? Considering the effects of neutrophils on macrophages and on cardiomyocytes, study of interactions with other cell types may have made more sense.

---

## [Author Response]

Essential Revisions:1) Figures 1, supplement 1 and Figure 3 supplement 1 show the variation in the percentage of monocyte subpopulations and neutrophil CD10neg over the course of 5 days after the onset of the symptoms. Showing absolute numbers would provide information about the possible mobilization of these cells from the bone marrow or other reservoirs as suggested by the authors.

We performed a time course analysis as suggested by the reviewers, and the results are reported in Figure 1—figure supplement 1 and Figure 3—figure supplement 1B. Variations of absolute numbers (i.e., number of cells/µL) calculated over the same time course showed similar pattern obtained using variations in percentage, confirming the conclusions drawn from previous data. The Results section was changed accordingly.

2) Do CD14+HLA-DRneg/low monocytes together with CD16+CD66b+CD10neg neutrophils improve ROC values compared to single parameters alone?

As suggested by the reviewers, we performed ROC curve analysis discriminating UA/STEMI patients combining CD14^+^HLA-DR^neg/low^ monocytes and CD16^+^CD66b^+^CD10^neg^ neutrophils. AUC resulted equal to 0.970 with 95% CI: 0.927-1, a higher value compared to those ones obtained by single parameters (0.949, 0.798 respectively). Regarding UA/NSTEMI, AUC of combination resulted equal to 0.772 with 95% CI: 0.584-0.960, a higher value compared to CD16^+^CD66b^+^CD10^neg^ neutrophils (0.687, 95% CI: 0.482-0.892) but lower compared to CD14^+^HLA-DR^neg/low^ monocytes (0.786, 95% CI: 0.612-0.961). Thus, the combination of two parameters improved the ROC curve providing maximal ROC-AUC for the ability to selectively discriminate STEMI versus UA patients. Graphical representation of the obtained ROC curves is shown in Author response image 1. The Results section was changed accordingly.

Supplemental note

Following a later check we have unfortunately to point out that a wrong graphical representation of CD14^+^HLA-DR^neg/low^ monocytes as percentage was loaded in the original submission (Figure 1—figure supplement 2B). We deeply apologize for this. As announced in the transparent reporting form presented at the time of the original submission we will provide the source data file.

3) CMV+ patients exhibit an increased prevalence of CD4^+^CD28null T cells. The authors show that in these patients CD4^+^CD28null T cells correlate with levels of IFNγ and high levels of neutrophils CD10neg in CMV+ positive patients, hence providing a link between these three parameters. However, mechanistically, immature neutrophils only stimulate CD4^+^CD28+ T cells to produce IFNγ. What is the relation then between CD4^+^CD28null T cells and immature neutrophils? What is the link the authors want to do here? These discrepancies should be discussed.

Our in vitro experiments showed that CD10^neg^ neutrophils had no effect on CD4^+^CD28^null^ T-cells (Figure 9A, 9B), thus demonstrating that overproduction of IFN-γ is confined to CD4^+^ T-cells expressing CD28. Moreover, using bioinformatic tools (PCA, hierarchical clustering, and Spearman-correlation matrix) we were able to highlight the tight relationship among the peripheral expansion of immature CD10^neg^ neutrophils, CMV-altered CD4^+^ T-cell homeostasis, CMV IgG titers, and high levels of IFN-γ in patients with large AMI. (Figure 8, Figure 8—figure supplement 1, Figure 8—figure supplement 2). As extensively discussed, these findings suggest that CD4^+^CD28^+^ T-cells from CMV^+^ patients display a distinct phenotype. Environmental factors especially CMV infection are likely to provide a significant contribution to functional diversity of CD4^+^ T-cells. Our results highlight that CD4^+^CD28^+^ T-cells from CMV-seropositive AMI patients are skewed toward a Th1 phenotype, producing large amounts of IFN-γ in presence of CD10^neg^ neutrophils.

4) Where do the authors think the interaction between neutrophils CD10neg and lymphocytes occurs? In the circulation? In the infarcted heart? Do the authors see increased IL-12 levels in plasma from patients with increased amounts of neutrophils CD10neg? Also, the authors should discuss this potential interaction.

A crucial role for neutrophils in the orchestration in adaptive immunity has recently emerged. Neutrophils can traffic to lymph nodes during homeostasis and inflammatory conditions. Our new data obtained from the mouse model of AMI are first to provide evidence that immature neutrophils can populate heart-draining lymph nodes in response to acute ischemic injury. Consistent with recent studies we found a small population of neutrophils in mediastinal lymph nodes at steady state (Figure 7A). In mice subjected to ischemia followed by 3 hours of reperfusion the percentage of neutrophils in heart-draining lymph nodes increased significantly (Figure 7A, 7B). Of interest, we found that approximately 50% of infiltrating neutrophils were CD101^neg^ cells. Morphologic analysis of cytospins confirmed the presence of neutrophils with ring‐shaped nuclei as well as mature neutrophils with segmented nuclei in mediastinal lymph nodes (Figure 7B). Therefore, it is intriguing to speculate that CD10^neg^ neutrophils can migrate into lymph nodes where they could encounter T-cells and shape adaptive immune responses. Future studies that employ intravital microscopy are required to visualize immature neutrophils dynamics mobilized from the bone marrow and recruited to lymph nodes after AMI.

Our additional bioinformatic analysis presented in the revised manuscript shows that circulating levels of IL-12 resulted enhanced in CMV^+^ patients with increased frequency of CD10^neg^ neutrophils (Figure 8—figure supplement 2B). Moreover, Spearman's correlation matrix showed multiple inter-correlations among IL-12 circulating levels, CMV IgG titers, frequency of CD10^neg^ neutrophils, expanded CD4^+^CD28^null^ T-cells and particularly circulating levels of IFN-γ (Figure 8—figure supplement 2C). The Results section was changed accordingly.

5) Is there only a relationship between CMV+ patients and CD4^+^CD28null? In Figure 7, supplement 1 panel C seems not to be a clear correlation between CMV positivity and neutrophil CD10neg numbers. Is this the case? For instance, with CMV-IgG? If not in numbers, how CMV positivity affects neutrophil CD10neg phenotype? Do neutrophil CD10neg isolated from patients CMV+ or CMV- have different ability to induce IFNγ in T-cells?

In the Results section of the original submission, we had suggested that cell expansion of CD10^neg^ cells may be due prevalently to grade of coronary disease masking the effect of CMV-seropositivity. To assess the questions raised by the reviewers we investigated the correlation between CD10^neg^ neutrophils and CMV-IgG titers as suggested. Principal component analysis (PCA) performed with the percentage of CD10^neg^ neutrophils, IFN-γ levels and CMV-IgG titers (Figure 8—figure supplement 2A) showed a better cluster grouping compared to that one obtained with percentage of CD4^+^CD28^null^ cells (Figure 8C), with similar represented total variance (PC1+PC2: 79.5% vs. 83% respectively). Moreover, as reported in the panel (Figure 8—figure supplement 2C), a highly significant positive correlation resulted between CMV-IgG and the percentage of CD4^+^CD28^null^ cells, whereas a non-significant correlation was observed between CMV-IgG and CD10^neg^ cells. In the heatmap shown in Figure 8—figure supplement 2B, CMV-IgG titers and the percentage of CD4^+^CD28^null^ cells are grouped together, confirming the relationship between CMV-seropositivity and the expansion of CD4^+^CD28^null^ cells. In addition, to test whether neutrophils from CMV^-^ or CMV^+^ patients have different ability to induce IFN-γ, CD4^+^ T-cells from CMV^+^ patients were cultured in presence of supernatants derived from CMV-seronegative patients (Figure 9—figure supplement 1). We found that CD10^neg^ neutrophil supernatants markedly enhanced IFN-γ secretion (Figure 9—figure supplement 1), thus suggesting that CD10^neg^ neutrophils from CMV^-^ and CMV^+^ patients exhibit the same ability to induce IFN-γ production by CD4^+^ T-cells.

6) The authors focus on the interaction of immature neutrophils and the adaptive immune system, but multiple studies associate neutrophil activity with monocyte/macrophage functionality. Is there any relation between CD10neg neutrophils and the observed phenotype of monocytes? Is the alarmin S100A9 relevant for monocyte chemotaxis?

In the original submission (Figure 6) a significant correlation between CD10^pos^ neutrophils and HLA-DR^neg/low^ monocytes and their link to circulating levels of immune-inflammation markers had been shown through Spearman-correlation matrix. Since a crucial role for immature neutrophils in the orchestration of adaptive immunity has recently emerged, in the present study we mostly focused on neutrophil-mediated regulation of T-cell response. However, to address the concerns raised by the reviewers we performed additional in vitro experiments investigating potential interactions/relations between CD10^neg^ neutrophils and CD14^+^HLA-DR^neg/low^ monocytes. Horckmans and co-workers showed that neutrophils orchestrate cardiac healing after AMI by influencing macrophage polarization. This work (Ref. 21) reported an induction of MerTK expression in macrophages (differentiated from peripheral blood monocytes of healthy donors) exposed to IL-4 in the presence of neutrophil supernatants from healthy individuals. Therefore, in the present study we assessed MerTK expression on macrophages differentiated from CD14^+^HLA-DR^neg/low^ and CD14^+^HLA-DR^high^ monocytes and polarized in the presence of supernatants derived from CD10^neg^ neutrophils as well as CD10^pos^ neutrophils (Figure 4—figure supplement 3C, 3D). CD14^+^HLA-DR^neg/low^/CD14^+^HLA-DR^high^ cells and CD10^neg^/CD10^pos^ neutrophils were isolated from patients with AMI. Of interest, we found a significantly higher expression of MerTK on macrophages differentiated from CD14^+^HLA-DR^neg/low^ monocytes and stimulated with IFN-γ and IL-4 in presence of secretome derived from CD10^pos^ neutrophils. Efficient clearance of apoptotic cells by human macrophages requires MerTK induction. Therefore, we speculated that HLA-DR^neg/low^ monocytes could have the potential to differentiate into macrophages with enhanced capacity to phagocyte apoptotic cells when exposed to CD10^pos^ neutrophils post-AMI. Nevertheless, in vitro studies have limited ability to explain the complexity of macrophage activation within the cardiac environment after ischemic injury.

Regarding the last question, we underline that alarmin S100A9 has been shown to have chemotactic activity for monocytes. Previous experiments have focused on monocytic THP-1 cells and monocytes isolated from healthy human subjects. To address the issue raised by the reviewers, we analyzed S100A9-induced chemotaxis of CD14^+^HLA-DR^neg/low^/CD14^+^HLA-DR^high^ monocytes from patients with AMI. We found that S100A9 significantly induced the migration of CD14^+^HLA-DR^high^ as well as CD14^+^HLA-DR^neg/low^ monocytes (Figure 4—figure supplement 3E). It is tempting to speculate that S100A9 regulates CD14^+^HLA-DR^neg/low^/CD14^+^HLA-DR^high^ monocytes recruitment during AMI by functioning as a chemoattractant.

7) There are major discrepancies between mouse and human immature monocytes and neutrophils. The authors should discuss it. The samples analyzed in mice for RNAseq are total FACS-sorted neutrophils. Do the authors see more similarities when analyzing CD101- mouse neutrophils?

To address the concern raised by the reviewers we performed additional experiments and isolated by FACS sorting CD11b^pos^Ly6G^pos^CD101^neg^ and CD11b^pos^Ly6G^pos^CD101^pos^ cells from peripheral blood of mice subjected to ischemia/reperfusion. We performed RT-qPCR of the highlighted genes in MA plot of Figure 4—figure supplement 1B. Of interest, we found that CD101^neg^ neutrophils express higher amounts of Irg1, Il1r1, Mmp8, Nos2, and Stat3 than CD101^pos^ neutrophils and showed markedly reduced CD101 expression (Figure 4—figure supplement 1D, 1E) in good agreement with RNAseq analysis. Our findings emphasize interspecies differences in immune responses after ischemic injury, suggesting caution in applying findings in murine models of myocardial ischemia/reperfusion to patients with AMI.

8) How are the levels of immature neutrophils altered in the bone marrow of mice after ischemia-reperfusion? The authors should also provide absolute numbers of CD101+ and CD101- neutrophils within the ischemic heart to evaluate their contribution.

To address the issues raised by the reviewers we performed additional experiments using the mouse model of reperfused AMI. As showed in the new Figure 4—figure supplement 2C the levels of CD101^neg^ neutrophils in the bone marrow of mice subjected to ischemia/reperfusion were markedly reduced, suggesting early release of immature neutrophils from bone marrow into the circulation after AMI. The absolute numbers of CD101^neg^ neutrophils as well as CD101^pos^ neutrophils within the ischemic heart at 3 hours and 24 hours after reperfusion were now reported in the Figure 6A, 6B.

9) In some cases, conclusions are not supported by robust data. For example, the authors conclude that CD14+HLA-DRneg/lo monocytes play a crucial role in post-infarction inflammation based exclusively on in vitro experiments. Moreover, conclusions regarding the pro-inflammatory role of immature neutrophils are based on in vitro data and associative studies. Please tone down the conclusions and discuss the limitations related to the absence of in vivo experiments depleting immature neutrophils.

We agree with the reviewers that in vitro studies have limited ability to explain the complexity of immunoregulatory mechanisms operating during post-infarction inflammation. We have now explicitly discussed the limitations in the revised manuscript to mitigate the specific claims, as suggested.

10) What is the fate of immature neutrophils in the infarct? These cells are typically short-lived. Did they rapidly undergo apoptosis? Did they mature rapidly within the infarct environment? The mouse model may allow the authors to address these questions.

As mentioned in the revised manuscript, studies taking advantage of intravital microscopy are needed to investigate the fate of immature neutrophils mobilized from the bone marrow and recruited to the ischemic myocardium. However, taking into account the questions raised by the reviewers, to further define the functional properties of the immature neutrophil populations derived from AMI patients we investigated spontaneous apoptosis. Remarkably, the apoptotic rate of CD10^neg^ neutrophils was strongly reduced compared with CD10^pos^ neutrophils after 24 hours in vitro culture as determined by flow cytometric analysis using Vybrant DyeCycle Violet/SYTOX AADvanced stain (Figure 4E) as well as Apotracker Green (Figure 4—figure supplement 3A). The decreased apoptosis rate of CD10^neg^ neutrophils was reflected by an increase in the percentage of live cells. Morphological assessment of cytospins (Figure 4—figure supplement 3B) confirmed the results obtained by flow cytometry. We also studied the apoptosis rate of CD101^neg^ and CD101^pos^ neutrophils isolated from ischemic hearts 24 hours after reperfusion. We found that CD101^neg^ neutrophils were more resistant to apoptosis than their CD101^pos^ counterparts (Figure 6D). Apoptosis is essential for neutrophil functional shutdown and effective resolution of inflammation requires timely phagocytic clearance of apoptotic neutrophils by macrophages. Therefore, it is tempting to speculate that apoptosis resistance of immature neutrophils may play a crucial role, hitherto overlooked, after MI and future studies will certainly shed light on its role in cardiac wound healing.

11) The authors need to explain the rationale for assessment of specific inflammatory genes. For example, in neutrophils, what was the basis for selection of these specific genes (MMP9, IL1R1, IL1R2, STAT3 etc), vs. other inflammatory genes known to be expressed at high levels by neutrophils?

Our first goal was to identify specific surface antigens that characterize normal-density CD10^neg^ neutrophils derived from patients with AMI. Therefore, we examined markers of neutrophil maturation/activation often investigated in inflammatory diseases. Amazingly, we found that CD11b, CD101, CD114 and CD177 were expressed at equivalent levels on CD10^neg^ neutrophils versus CD10^pos^ neutrophils (Figure 4C). In the absence of significance, these data were not included in the original submission. Next, we investigated the expression of inflammatory mediators playing a pathophysiological role in wound healing after AMI expressed at high levels by neutrophils (MMP-9, S100A9) and, above all regulated in circulating neutrophils as well as in infarct neutrophils in the mouse model of reperfused AMI (Figure 4—figure supplement 1).

12) The rationale for the use of CD101 as a marker differentiating between mature and immature neutrophils in the mouse needs to be better presented. It seems that the authors identified CD101 as a neutrophil gene that is downregulated following ischemia. On that basis, they used CD101 as a maturation marker. However, why does this suggest that CD101 selectively labels mature vs immature neutrophils? Perhaps its downregulation reflects neutrophil activation, rather than immaturity. Perhaps the use of CD101 as a marker for mature neutrophils in reference 14 may provide a better rationale.

As reminded by the reviewers using next-generation RNA sequencing we identified Cd101 among the genes down-regulated by ischemia in circulating neutrophils (Figure 4—figure supplement 1B). To address the concern raised by the reviewers we performed additional experiments. As reported in the (Figure 4—figure supplement 1D) we found that CD11b^pos^Ly6G^pos^CD101^neg^ cells sorted from peripheral blood after AMI showed markedly reduced expression of Cd101. However, as suggested by the reviewers in the revised manuscript we refer CD101 as a surface marker to identify the mature neutrophil subset among the heterogeneous neutrophil populations, released into the bloodstream after ischemia/reperfusion. Anyway, as revealed by morphological analysis circulating CD101^pos^ neutrophils have a mature morphology, whereas CD101^neg^ neutrophils are immature neutrophils with ring‐shaped nuclei (Figure 4—figure supplement 2B). More importantly, analysis of cytospin preparations of sorted neutrophils also showed that CD101^neg^ neutrophils isolated from ischemic myocardium 24 hours after reperfusion still have an immature morphology (Figure 6C).

13) The rationale for the study of CMV seropositivity is not well-explained. The data show infiltration of the infarcted heart with immature neutrophils and CD14+HLA-DRneg monocytes. One would have anticipated experiments investigating the (proposed) role of these cells in the post-infarction inflammatory response. The focus on CMV+ patients needs to be better introduced in the manuscript.

The study of CMV-seropositivity was not one of our primarly goal(s) but was introduced during the course of the study following our attempt to explain the anomalous, very skewed distributions of IFN-γ levels in patients with AMI having expanded CD10^neg^ neutrophils and increased frequency of CD4^+^CD28^null^ T-cells (Figure 8B). As mounting evidence indicated that altered T-cell homeostasis and increased frequency of circulating CD28^null^ T-cells are linked to cytomegalovirus (CMV) seropositivity, we next analyzed the impact of CMV serostatus on CD4^+^CD28^null^ T-cells frequency. Strikingly, we discovered that CD4^+^CD28^null^ expansion occurs in cytomegalovirus-seropositive patients (Figure 8D) and that maximum levels of circulating IFN-γ among ACS patients were detected in CMV-seropositive STEMI patients displaying increased levels of CD10^neg^ neutrophils (Figure 8E). Therefore, we searched for criteria to define expansion of CD10^neg^ neutrophils and CD4^+^CD28^null^ T-cell frequency using CMV-seropositivity as discriminating factor (Figure 8—figure supplement 1A-D). Derived cut-offs permitted grouping patients into well-defined clusters by PCA (Figure 8C) but chiefly providing evidence of the above-described link (Figure 8—figure supplement 1E). Moreover, derived cut-off for CD10^neg^ neutrophil frequency together with CMV-seropositivity factor (±) permitted also to highlight the relationship among CD10^neg^, IL-12, IFN-γ as described in response to point 4 (Figure 8—figure supplement 2B). The validity to introduce CMV-seropositivity in the study is strengthened looking at PCA in Figure 8—figure supplement 2A, since patients resulted better clustered compared to PCA where CD4^+^CD28^null^ T-cell frequency were analyzed in place of CMV IgG titers, this with a negligible loss of total explained variance. We have now explicitly highlighted these points in the revised manuscript.

Facit: operative steps over the course of the study conducted us to consider CMV positivity as factor strictly jointed to altered frequency of CD4^+^CD28^null^ T-cells associated with IFN-γ production and in turn of CD10^neg^ neutrophils and to represent their complex interplay through bioinformatic tools.

14) What was the rationale for the experiment examining interactions of CD10neg neutrophils with T cells? Considering the effects of neutrophils on macrophages and on cardiomyocytes, study of interactions with other cell types may have made more sense.

Increasing evidence supports that under pathological conditions low-density neutrophils display both immunosuppressive or proinflammatory functions. However, the immunoregulatory effects of normal-density CD10^neg^ neutrophils after AMI and in context of cardiovascular diseases remains largely unexplored. Since a crucial role for immature neutrophils in the orchestration of adaptive immunity has recently emerged, we mostly focused on neutrophil-mediated regulation of T-cell response. However, to address the concerns raised by the reviewers (as described above in response to point 6) we performed additional in vitro experiments investigating potential interactions between CD14^+^HLA-DR^neg/low^/CD14^+^HLA-DR^high^ monocytes and CD10^neg^/CD10^pos^ neutrophils (Figure 4—figure supplement 3C, 3D).